# Self-Aware Personalized Federated Learning

**Huili Chen, Jie Ding, Eric Tramel, Shuang Wu, Anit Kumar Sahu, Salman Avestimehr, Tao Zhang**

Alexa AI, Amazon

`{chehuili,jiedi,eritrame,wushuan,anitsah,avestime,taozhng}@amazon.com`

## Abstract

In the context of personalized federated learning (FL), the critical challenge is to balance local model improvement and global model tuning when the personal and global objectives may not be exactly aligned. Inspired by Bayesian hierarchical models, we develop a self-aware personalized FL method where each client can automatically balance the training of its local personal model and the global model that implicitly contributes to other clients' training. Such a balance is derived from the *inter-client and intra-client uncertainty quantification*. A larger inter-client variation implies more personalization is needed. Correspondingly, our method uses *uncertainty-driven local training steps and aggregation rule* instead of conventional local fine-tuning and sample size-based aggregation. With experimental studies on synthetic data, Amazon Alexa audio data, and public datasets such as MNIST, FEMNIST and Sent140, we show that our proposed method can achieve significantly improved personalization performance compared with the existing counterparts.

## 1 Introduction

Federated learning (FL) (Konevcny et al., 2016; McMahan et al., 2017) is transforming machine learning (ML) ecosystems from "centralized in-the-cloud" to "distributed across-clients," to potentially leverage the computation and data resources of billions of edge devices (Lim et al., 2020), without raw data leaving the devices. As a distributed ML framework, FL aims to train a global model that aggregates gradients or model updates from the participating edge devices. Recent research in FL has significantly extended its original scope to address the emerging concern of personalization, a broad term that often refers to an FL system that accommodates client-specific data distributions of interest (Ding et al., 2022).

In particular, each client in a personalized FL system holds data that can be potentially non-identically and independently distributed (non-IID). For example, smart edge devices at different houses may collect audio data (Purington et al., 2017) of heterogeneous nature due to, e.g., accents, background noises, and house structures. Each device hopes to improve its on-device model through personalized FL without transmitting sensitive data.

While the practical benefits of personalization have been widely acknowledged, its theoretical understanding remains unclear. Existing works on personalized FL often derive algorithms based on a pre-specified optimization formulation, but explanations regarding the formulation or its tuning parameters rarely go beyond heuristic statements.

In this work, we take a different approach. Instead of specifying an optimization problem to solve, we start with a toy example and develop insights into the nature of personalization from a statistical uncertainty perspective. We aim to answer the following critical questions regarding personalized FL.

*(Q1) The lower-bound baselines of personalized FL can be obtained in two cases, i.e., each client performs local training without FL, or all clients participate in conventional FL training. However, the upper-bound for the client is unclear.*

36th Conference on Neural Information Processing Systems (NeurIPS 2022).

*(Q2) Suppose that the goal of each client is to improve its local model performance. How to design an FL training that interpret the global model, suitably aggregate local models and fine-tune each client's local training automatically?*

Both questions Q1 and Q2 are quite challenging. Q1 demands a systematic way to characterize the client-specific and globally-shared information. Such a characterization is agnostic to any particular training process being used. To this end, instead of studying personalized data in full generality, we restrict our attention to a simplified and analytically tractable setting: *two-level Bayesian hierarchical models*, where the top and bottom level describes inter-client and intra-client uncertainty, respectively.

The above Q2 requires FL updates to be adaptive to the nature of the underlying discrepancy among client-specific local data, including sample sizes and distributions. The popular aggregation approach that uses a sample size-based weighting mechanism (McMahan et al., 2017) does not account for the distribution discrepancy. Meanwhile, since clients' data is not shared, estimating the underlying data distributions is unrealistic. Consequently, it is highly nontrivial to measure and calibrate such a discrepancy in FL settings. Addressing the above issues is a key motivation of our work.

## 1.1 Contributions

We make the following technical contributions:

- Interpreting personalization from a *hierarchical model-based* perspective and providing theoretical analyses for FL training.
- Proposing Self-FL, an active personalized FL solution that guides local training and global aggregation via inter- and intra-client *uncertainty quantification*.
- Presenting a novel implementation of Self-FL for deep learning, consisting of automated hyper-parameter tuning for clients and an adaptive aggregation rule.
- Evaluating Self-FL on Sent140 and Amazon Alexa audio data. Empirical results show promising personalization performance compared with existing methods.

To our best knowledge, Self-FL is the first work that connects personalized FL with hierarchical modeling and utilizes uncertainty quantification to drive personalization.

## 1.2 Related work

**Personalized FL.** The term *personalization* in FL often refers to the development of client-specific model parameters for a given model architecture. In this context, each client aims to obtain a local model that has desirable test performance on its local data distribution. Personalized FL is critical for applications that involve statistical heterogeneity among clients.

A research trend is to adapt the global model for accommodating personalized local models. To this end, prior works often integrate FL with other frameworks such as multi-task learning (Smith et al., 2017), meta-learning (Jiang et al., 2019; Khodak et al., 2019; Fallah et al., 2020a;b; Al-Shedivat et al., 2021), transfer learning (Wang et al., 2019; Mansour et al., 2020), knowledge distillation (Li and Wang, 2019), and lottery ticket hypothesis (Li et al., 2020a). For example, DITTO (Li et al., 2021) formulates personalized FL as a multi-task learning (MTL) problem and regularizes the discrepancy of the local models to the global model using the $\ell_2$ norm. FedEM Marfoq et al. (2021) uses the MTL formulation for personalization under a finite-mixture model assumption and provides federated EM-like algorithms. We refer to Vanhaesebrouck et al. (2017); Hanzely et al. (2020); Huang et al. (2021) for more MTL-based approaches and theoretical bounds on the optimization problem. From the perspective of meta-learning, pFedMe (Dinh et al., 2020) formulates a bi-level optimization problem for personalized FL and introduces Moreau envelopes as clients' regularized loss. PerFedAvg (Fallah et al., 2020b) proposes a method to find a proper initial global model that allows a quick adaptation to local data. pFedHN Shamsian et al. (2021) proposes to train a central hypernetwork for generating personalized models. FedFOMO Zhang et al. (2020) introduces a method to compute the optimally weighted model aggregation for personalization by characterizing the contribution of other models to one client. Later on, we will focus on the experimental comparison of Self-FL with DITTO, pFedMe, and PerFedAvg, since they represent two popular personalized FL formulations, namely multi-task learning and meta-learning.

A limitation of existing personalized FL methods is that they do not provide a clear answer to question Q1. Consequently, it is unclear how to interpret the FL-trained results. For example, in the extreme case where the clients are equipped with irrelevant tasks and data, any personalized FL methods that require a pre-specified global objective (e.g., Ditto (Li et al., 2021), pFedMe (Dinh et al., 2020)) may cause significant biases to local clients.

## 2 Bayesian Perspective of Personalized FL

We discuss how Self-FL approaches personalized FL with theoretical insights from the Bayesian perspective in this section. The notations are defined as follows. Let $\mathcal{N}(\mu, \sigma^2)$ denote Gaussian distribution with mean $\mu$ and variance $\sigma^2$. For a positive integer $M$, let $[M]$ denote the set $\{1, \ldots, M\}$. Let $\sum_{m \neq i}$ denote the summation over all $m \in [M]$ except for $m = i$. We summarize frequently used notation in Table 3 of Appendix and the derivation for general parametric models in Section D.

### 2.1 Understanding personalized FL through a two-level Gaussian model

To develop insights, we first restrict our attention to the following simplified case. Suppose that there are $M$ clients. From the server's perspective, it is postulated that data $z_1, \ldots, z_M$ are generated from the following two-layer Bayesian hierarchical model:

$$\theta_m \mid \theta_0 \overset{\text{IID}}{\sim} \mathcal{N}(\theta_0, \sigma_0^2), \quad z_m \mid \theta_m \overset{\text{IID}}{\sim} \mathcal{N}(\theta_m, \sigma_m^2), \tag{1}$$

for all clients with indices $m = 1, \ldots, M$. Here, $\sigma_0^2$ is a constant, and $\theta_0$ is a hyperparameter with a non-informative flat prior (denoted by $\pi_0$). The above model represents both the connections and heterogeneity across clients. In particular, each client's data are distributed according to a client-specific parameter ($\theta_m$), which follows a distribution decided by a parent parameter ($\theta_0$). The parent parameter is interpreted as the root of shared information. In the rest of the section, we often study client 1's local model as parameterized by $\theta_1$ without loss of generality. Under the above model assumption, the parent parameter $\theta_0$ that represents the global model has a posterior distribution $p(\theta_0 \mid z_{1:M}) \sim \mathcal{N}(\theta^{(\text{G})}, v^{(\text{G})})$, where:

$$\theta^{(\text{G})} \triangleq \frac{\sum_{m \in [M]} (\sigma_0^2 + \sigma_m^2)^{-1} z_m}{\sum_{m \in [M]} (\sigma_0^2 + \sigma_m^2)^{-1}}, \quad v^{(\text{G})} \triangleq \frac{1}{\sum_{m \in [M]} (\sigma_0^2 + \sigma_m^2)^{-1}}. \tag{2}$$

The $z_m$ in Eqn. (2) may also be denoted by $\theta_m^{(L)}$, for reasons that will be seen in Eqn. (3). From the perspective of client $m$, we suppose that the postulated model is Eqn. (1) for $m = 2, \ldots, M$, and $\theta_1 = \theta_0$. It can be verified that the posterior distributions of $\theta_1$ without and with global Bayesian learning are $p(\theta_1 \mid z_1) \sim \mathcal{N}(\theta_1^{(L)}, v_1^{(L)})$ and $p(\theta_1 \mid z_{1:M}) \sim \mathcal{N}(\theta_1^{(\text{FL})}, v_1^{(\text{FL})})$, respectively, which can be computed as:

$$\theta_1^{(L)} \triangleq z_1, \quad v_1^{(L)} \triangleq \sigma_1^2,$$
$$\theta_1^{(\text{FL})} \triangleq \frac{\sigma_1^{-2} \theta_1^{(L)} + \sum_{m \neq 1} (\sigma_0^2 + \sigma_m^2)^{-1} \theta_m^{(L)}}{\sigma_1^{-2} + \sum_{m \neq 1} (\sigma_0^2 + \sigma_m^2)^{-1}}, \tag{3}$$
$$v_1^{(\text{FL})} \triangleq \frac{1}{\sigma_1^{-2} + \sum_{m \neq 1} (\sigma_0^2 + \sigma_m^2)^{-1}}.$$

The first and the second distribution above describe the learned result of client 1 from its local data and from all clients' data in hindsight, respectively. Using mean square error as risk, the Bayes estimate of $\theta_1$ or $\theta_0$ is the mean of the posterior distribution, namely $\theta_1^{(L)}$ and $\theta_1^{(\text{FL})}$ as defined above.

The flat prior on $\theta_0$ can be replaced with any other distribution to bake prior knowledge into the calculation. We consider the flat prior because the knowledge of the shared model is often vague in practice. The above posterior mean $\theta_1^{(\text{FL})}$ can be regarded as the optimal point estimation of $\theta_1$ given all the clients' data, thus is referred to as "FL-optimal". $\theta^{(\text{G})}$ can be regarded as the "global-optimal." The posterior variance quantifies the reduced uncertainty conditional on other clients' data. Specifically, we define the following *Personalized FL gain* for client 1 as:

$$\text{GAIN}_1 \triangleq \frac{v_1^{(L)}}{v_1^{(\text{FL})}} = 1 + \sigma_1^2 \sum_{m \neq 1} (\sigma_0^2 + \sigma_m^2)^{-1}.$$

*Remark* 2.1 (**Interpretations of the posterior quantities**). Each client, say client 1, aims to learn $\theta_1$ in the personalized FL context. Its learned information regarding $\theta_1$ is represented by the Bayesian posterior of $\theta_1$ conditional on either its local data $z_1$ (without communications with others), or the data $z_{1:M}$ in hindsight (with communications). For the former case, the posterior uncertainty described by $v_1^{(L)}$ depends only on the local data quality $\sigma_1^2$. For the latter case, the posterior mean $\theta_1^{(\text{FL})}$ is a weighted sum of clients' local posterior means, and the uncertainty will be reduced by a factor of $\text{GAIN}_1$. Since a point estimation of $\theta_1$ is of particular interest in practical implementations, we treat $\theta_1^{(\text{FL})}$ as the theoretical limit in the FL context (recall question Q1). To develop further insights into $\theta_1^{(\text{FL})}$, we consider the following extreme cases.

- As $\sigma_0^2 \to \infty$, meaning that the clients are barely connected, the quantities $\theta_1^{(\text{FL})}$ and $v_1^{(\text{FL})}$ reduce to $\theta_1^{(\text{L})}$ and $v_1^{(\text{L})}$, respectively; meanwhile, the personalized FL gain approaches one, the global parameter mean $\theta^{(\text{G})}$ becomes a simple average of $\theta_m^{(\text{L})}$ (or $z_m$), and the global parameter has a large variance.

- When $\sigma_0^2 = 0$, meaning that the clients follow the same underlying data-generating process and the personalized FL becomes a standard FL, we have $\theta_1^{(\text{FL})} = \theta^{(\text{G})}$, which is a weighted sum of clients' local optimal solution with weight proportional to $\sigma_m^{-2}$ (namely client $m$'s precision).

- When $\sigma_1^2$ is much smaller than all other $\sigma_m^2$'s and $\sigma_0^2$, and $M$ is not too large, meaning that client 1 has much higher quality data compared with the other clients combined, we have $\theta_1^{(\text{FL})} \approx z_1 = \theta_1^{(\text{L})}$ and $\text{GAIN}_1 \approx 1$. In other words, client 1 almost learns on its own. Meanwhile, client 1 can still contribute to other clients through the globally shared parameter $\theta_0$. For example, the gain for client 2 would be $\text{GAIN}_2 \geq \sigma_2^2/\sigma_1^2$, which is much larger than one.

*Remark* 2.2 (**Local training steps to achieve** $\theta_1^{(\text{FL})}$). Suppose that client 1 performs $\ell$ training steps using its local data and negative log-likelihood loss. We show that with a suitable number of steps and initial value, client 1 can obtain the intended $\theta_1^{(\text{FL})}$. The local objective is:

$$\theta \mapsto (\theta - z_1)^2/(2\sigma_1^2) = (\theta - \theta_1^{(\text{L})})^2/(2\sigma_1^2), \tag{4}$$

which coincides with the quadratic loss. Let $\eta \in (0, 1)$ denote the learning rate. By running the gradient descent:

$$\theta_1^\ell \leftarrow \theta_1^{\ell-1} - \eta \frac{\partial}{\partial \theta}\left( (\theta - \theta_1^{(\text{L})})^2/(2\sigma_1^2) \right)|_{\theta_1^{\ell-1}}$$
$$= \theta_1^{\ell-1} - \eta(\theta_1^{\ell-1} - \theta_1^{(\text{L})})/\sigma_1^2 \tag{5}$$

for $\ell$ steps with the initial value $\theta_1^{\text{INIT}}$, client 1 will obtain:

$$\theta_1^\ell = \left(1 - (1 - \sigma_1^{-2}\eta)^\ell\right)\theta_1^{(\text{L})} + (1 - \sigma_1^{-2}\eta)^\ell \theta_1^{\text{INIT}}. \tag{6}$$

It can be verified that Eqn. (6) becomes $\theta_1^{(\text{FL})}$ in Eqn. (3) if and only if:

$$\theta_1^{\text{INIT}} = \frac{\sum_{m \neq 1}(\sigma_0^2 + \sigma_m^2)^{-1}\theta_m^{(\text{L})}}{\sum_{m \neq 1}(\sigma_0^2 + \sigma_m^2)^{-1}}, \tag{7}$$

$$(1 - \sigma_1^{-2}\eta)^\ell = \frac{\sum_{m \neq 1}(\sigma_0^2 + \sigma_m^2)^{-1}}{\sigma_1^{-2} + \sum_{m \neq 1}(\sigma_0^2 + \sigma_m^2)^{-1}}. \tag{8}$$

In other words, with a suitably chosen initial value $\theta_1^{\text{INIT}}$, learning rate $\eta$, and the number of (early-stop) steps $\ell$, client 1 can obtain the desired $\theta_1^{(\text{FL})}$.

## 3 Proposed Solution for Personalized FL

Our proposed Self-FL framework has three key components as detailed in this section: (i) proper initialization for local clients at each round, (ii) automatic determination of the local training steps, (iii) discrepancy-aware aggregation rule for the global model. These components are interconnected and contribute together to Self-FL's effectiveness. Note that points (i) and (iii) direct Self-FL to the regions that benefit personalization in the optimization space during local training, which is not considered in prior works such as DITTO Li et al. (2021) and pFedMe Dinh et al. (2020). Therefore, Self-FL is more than imposing implicit regularization via early stopping.

### 3.1 From posterior quantities to FL updating rules

In this section, we show how the posterior quantities of interest in Section 2 can be connected with FL, where clients' parameters suitably deviate from the global parameter, and the global parameter is a proper aggregation of clients' parameters. For notional brevity, we still consider the two-level Gaussian model in Subsection 2.1. For regular parametric models, we mentioned in Subsection D that one may treat $z_m$ as the local optimal solution $\theta_m^{(\text{L})}$, and its variance $\sigma_m^2$ as the variance of $\theta_m^{(\text{L})}$ due to finite sample.

Recall that each client $m$ can obtain the FL-optimal solution $\theta_m^{(\text{FL})}$ with the initial value $\theta_m^{\text{INIT}}$ in Eqn. (7) and tuning parameters $\eta, \ell$ in Eqn. (8). Also, it can be shown that $\theta_m^{\text{INIT}}$ is connected with the global-optimal $\theta^{(\text{G})}$ defined in Eqn. (2) through:

$$\theta_m^{\text{INIT}} = \theta^{(\text{G})} - \frac{(\sigma_0^2 + \sigma_m^2)^{-1}}{\sum_{k:\, k \neq m}(\sigma_0^2 + \sigma_k^2)^{-1}}(\theta_m^{(\text{L})} - \theta^{(\text{G})}). \tag{9}$$

The initial value $\theta_m^{\text{INIT}}$ in Eqn. (9) is unknown during training since $\theta_m^{(\text{L})}, \theta^{(\text{G})}$ are unknown. A natural solution is to update $\theta_m^{\text{INIT}}, \theta_m^{(\text{L})}$, and $\theta^{(\text{G})}$ iteratively, leading to the following personalized FL rule.

**Generic Self-FL**: At the $t$-th ($t = 1, 2, \ldots$) round of FL:

• *Each client $m$ receives the latest global model $\theta^{t-1}$ from the server and calculates:*

$$\theta_m^{t,\text{INIT}} \triangleq \theta^{t-1} - \frac{(\sigma_0^2 + \sigma_m^2)^{-1}}{\sum_{k:\, k \neq m}(\sigma_0^2 + \sigma_k^2)^{-1}}(\theta_m^{t-1} - \theta^{t-1}), \tag{10}$$

where $\theta_m^{t-1}$ is client $m$'s latest personal parameter at round $t-1$, initialized to be $\theta^0$. Starting from the above $\theta_m^{t,\text{INIT}}$, client $m$ performs gradient descent-based local updates with optimization parameters following Eqn. (8) or its approximations, and obtains a personal parameter $\theta_m^t$.

• *Server collects $\theta_m^t$, $m = 1, \ldots, M$, and calculates:*

$$\theta^t \triangleq \frac{\sum_{m \in [M]}(\sigma_0^2 + \sigma_m^2)^{-1}\theta_m^t}{\sum_{m \in [M]}(\sigma_0^2 + \sigma_m^2)^{-1}}. \tag{11}$$

In general, the above $\sigma_0^2, \sigma_m^2$ represent "inter-client uncertainty" and "intra-client uncertainty," respectively. When $\sigma_0^2$ and $\sigma_m^2$'s are unknown, they can be approximated using asymptotics of M-estimators or using practical finite-sample approximations (elaborated in Subsection 3.2).

We provide a theoretical understanding of the convergence. Consider the data-generating process that $\theta_m^{(\text{L})} \sim \mathcal{N}(\theta_m, \sigma_m^2)$ are independent, and $\theta_m \overset{\text{IID}}{\sim} \mathcal{N}(\theta_0, \sigma_0^2)$ where $\sigma_0^2$ is a fixed constant and each $\sigma_m^2$ may or may not depend on $M$. Let $O_p$ denote the standard stochastic boundedness. The following result gives an error bound of each client's personalized parameter and the server's parameter.

**Proposition 3.1.** *Assume that $\max_{m \in [M]} \sigma_m^2$ is upper bounded by a constant, and there exists a constant $q \in (0, 1)$ such that:*

$$\max_{m \in [M]} \frac{\sum_{k:\, k \neq m}(\sigma_k^2 + \sigma_0^2)^{-1}}{\sigma_m^{-2} + \sum_{k:\, k \neq m}(\sigma_k^2 + \sigma_0^2)^{-1}} \leq q. \tag{12}$$

*Suppose that at $t = 0$, the gap between the initial parameter and each client's FL-optimal value satisfies $|\hat{\theta}^0 - \theta_m^{(\text{FL})}| \leq C$ for all $m \in [M]$. Then, for every positive integer $t$, the quantities $\max_{m \in [1:M]} |\theta_m^t - \theta_m^{(\text{FL})}|, |\theta^t - \theta^{(\text{G})}|$ are both upper bounded by $C \cdot q^t + O_p(M^{-1/2})$ as $M \to \infty$.*

*Remark* 3.2 (*Interpretation of Proposition 3.1*). The proposition shows that the estimation error of each personalized parameter $\theta_m^t$ and the server parameter $\theta^t$ can uniformly go to zero as the number of clients $M$ and the FL round $t$ go to infinity. The error bound involves two terms. The first term $q^t$ corresponds to the optimization error. Typically, every $\sigma_m$ is a small value. If each client has a sample size of $N$, $q$ will be at the order of $M/(N+M)$. The second term $O_p(M^{-1/2})$ is a statistical error that vanishes as more clients participate. Intuitively, the initial model of each client and the global model at each round are averaged over many clients, so a larger pool leads to a smaller bias. The proof is nontrivial because the averaged terms are statistically dependent. For the particular case that each $\sigma_m^2$ is at the order of $N^{-1}$, we can see that the error is tiny when both $M$ and $N/M$ grow.

## 3.2  SGD-based practical algorithm for deep learning

For the training method proposed in Subsection 3.1, the quantities $\sigma_0^2$ and $\sigma_m^2$ are crucial as they affect the choice of learning rate $\eta_m$ and the early-stop rule. For many complex learning models, we do not know $\sigma_0^2$ and $\sigma_m^2$, and the asymptotic approximation of $\sigma_0^2$ and $\sigma_m^2$'s may not be valid due to a lack of regularity conditions. Furthermore, one often uses SGD instead of GD to perform local updates, so the stop rule depends on the batch size. In this subsection, we consider the above aspects and develop a practical algorithm for deep learning. Following the discussions in Subection D, we

can generally treat $\sigma_m^2$ as "uncertainty of the local optimal solution $\theta_m^{(\mathrm{L})}$ of client $m$", and $\sigma_0^2$ as "uncertainty of clients' underlying parameters." We propose a way to approximate them.

Assume that for each client $m$, we had $u$ independent samples of its data and the corresponding local optimal parameter $\theta_{m,1}, \ldots, \theta_{m,u}$. We could then estimate $\sigma_m^2$ by their sample variance. In practice, we can treat each round of local training as the optimization from a bootstrapped sample. In other words, at the round $t$, let client $m$ run sufficiently many steps (not limited by our tuning parameter $\ell$) until it approximately converges to a local minimum, denoted by $\theta_{m,t}$. To save computational costs, a client does not necessarily wait for its local convergence. Instead, we recommend that each client use its personal parameter as a surrogate of $\theta_{m,t}$ at each round $t$, namely to use $\theta_m^t$. In other words, at round $t$, we approximate $\sigma_m^2$ with:

$$\widehat{\sigma_m^2} = \text{empirical variance of } \{\theta_m^1, \ldots, \theta_m^t\}. \tag{13}$$

Likewise, at round $t$, we estimate $\sigma_0^2$ by:

$$\widehat{\sigma_0^2} = \text{empirical variance of } \{\theta_1^t, \ldots, \theta_M^t\}. \tag{14}$$

For multi-dimensional parameters, a counterpart of the derivations in earlier sections will involve matrix multiplications, which does not fit the usual SGD-based learning process. We thus simplify the problem by introducing the following uncertainty measures. For vectors $x_1, \ldots, x_M$, their empirical variance is defined as the trace of $\sum_{m \in [M]} (x_m - \bar{x})(x_m - \bar{x})^\mathsf{T}$, which is the sum of entry-wise empirical variances. $\widehat{\sigma_m^2}$ and $\widehat{\sigma_0^2}$ will be defined from such empirical variances, similar to Eqn. (13) and (14). In practice, for large neural network models, we suggest deploying Self-FL after the baseline FL training runs for certain rounds (namely warm-start) so that the fluctuation of local models does not hinder variance approximation.

Combining the above discussion and the generic method in Subsection 3.1, we propose our personalized FL solution in Algorithm 1. We include the implementation details of Self-FL in Section 4 and Appendix. We derived how to learn $\theta^{(\mathrm{G})}$ and $\theta_m^{(\mathrm{FL})}$ for parametric models in Subsection D. The key motivation of Algorithm 1 is that a client often cannot obtain the exact $\theta_m^{(\mathrm{L})}$ (especially in DL), so we interweave local training and personalization through an iterative FL procedure. Note that the computation of $\theta^{(\mathrm{G})}$ and $\theta_m^{(\mathrm{FL})}$ requires uncertainty measurements approximated from communications between the clients and the server. In the case where a client does not have sufficient labeled data, this client's intra-client uncertainty value $\sigma_m^2$ tends to be large and the local training step $l$ computed via Equation (8) will be small. This means that Self-FL will suggest a client with limited labeled data train only a few epochs on their local dataset, thus alleviating over-fitting.

---

**Algorithm 1** Self-aware Personal FL (Self-FL)

---

**Input:** A server and $M$ clients. Communication rounds $T$, client activity rate $C$, client $m$'s local data $D_m$ and learning rate $\eta_m$.
**for** each communication round $t = 1, \ldots T$ **do**
    Sample clients: $\mathbb{M}_t \leftarrow \max(\lfloor C \cdot M \rfloor, 1)$
    **for** each client $m \in \mathbb{M}_t$ **in parallel do**
        Distribute server model $\theta^{t-1}$ to client $m$
        Estimate $\widehat{\sigma_m^2}$ using Eqn. (13)
        Compute local step $l_m$ from Eqn. (8) and local initialization $\theta_m^{\mathrm{INIT}}$ via Eqn. (7)
        $\theta_m^t \leftarrow LocalTrain(\theta_m^{\mathrm{INIT}}, \eta_m, l_m; D_m)$
    Server estimates $\widehat{\sigma_0^2}$ using Eqn. (14)
    Server performs global aggregation and updates global model $\theta^t$ via Eqn. (11)

---

**Discussion.** The proposed Self-FL framework (described in Alg. 1) requires the selected clients to send their updated local models $\theta_m^t$ and the intra-client uncertainty values $\sigma_m^2$ to the server. After uncertainty-aware model aggregation, the server distributes the updated global model and the inter-client uncertainty $\sigma_0^2$ to the clients. Therefore, the communication cost of Self-FL is the same level as the standard FedAvg except for the additional cost of transmitting the uncertainty values (which are scalars). Compared to baseline FedAvg, Self-FL requires additional computation to obtain the inter-client and intra-client uncertainties via sample variance as shown in Eqn. (13) and (14). The extra computation cost of Self-FL is small since these sample variances are updated in an online manner as detailed in Appendix Subsection E. It is worth noting that our proposed personalization method for balancing local and global training in FL is applicable to other tasks involving auxiliary data since Self-FL does not depend on a particular formulation of the empirical risk.

# 4 Experimental Studies

## 4.1 Experimental setup

We use AWS p316 instances for all experiments. To evaluate the performance of Self-FL, we consider synthetic data, images, texts, and audios. The loss in Alg. 1 can be general. For our experiments on MNIST, FEMNIST, CIFAR10, Sent140, and wake-word detection, we used the cross-entropy loss. Our empirical results on DL problems suggest that Self-FL provides promising personalization performance in complex scenarios.

**Synthetic Data.** We construct a synthetic dataset based on the two-layer Bayesian model discussed in Subsection 2.1 where each data point is a scalar. The FL system has a total of $M = 20$ clients, and the task is to estimate the global and local model parameters. The heterogeneity level of the data can be controlled by specifying the inter-client uncertainty $\sigma_0^2$ and the local dataset size $N_m$ for each client $m$. We construct two synthetic FL datasets, one with high homogeneity and one with high heterogeneity, to investigate the performance of FL algorithms in different scenarios. The empirical results are provided in Appendix Subsection G.1.

**MNIST Data.** This image dataset has 10 output classes and input dimension of $28 \times 28$. We use a multinomial logistic regression model for this task. The FL system has a total of $1,000$ clients. We use the same non-IID MNIST dataset as FedProx (Li, 2020), where each client has samples of two-digit classes, and the local sample size of clients follows a power law (Li et al., 2020b).

**FEMNIST Data.** The federated extended MNIST (FEMNIST) dataset has 62 output classes. The FL system has $M = 200$ clients. We use the same approach as FedProx (Li et al., 2020b) for heterogeneous data generation where ten lower-case characters ('a'-'j') from EMNIST are selected, and each client holds five classes of them.

**CIFAR10 Data.** This image dataset has images of dimension $32 \times 32 \times 3$ and 10 output classes. We use the non-i.i.d. data provided by the pFedMe repository Canh T. Dinh (2020). There are a total of 20 clients in the system where each user has images of three classes.

**Sent140 Data.** This is a text sentiment analysis dataset containing tweets from Sentiment140 Go et al. (2009). It has two output classes and 772 clients. We generate non-i.i.d. data using the same procedure as FedProx Li (2020). Detailed setup and our experimental results on Sent140 are provided in Appendix Subsection G.3.

**Private Wake-word Data.** We also evaluate Self-FL on a private audio data from Amazon Alexa for the wake-word detection task. This is a common task for smart home devices where the on-device model needs to determine whether the user says a pre-specified keyword to wake up the device. This task has two output classes, namely 'wake-word detected' and 'wake-word absent.' The audio stream from the speaker is represented as a spectrogram and is fed into a convolution neural network (CNN). The CNN is a stack of convolutional and max-pooling layers (11 in total). The number of trainable parameters is around $3 \cdot 10^6$. The heterogeneity between clients comes from the intrinsic property of different device types (e.g., determined by hardware and use scenarios). We use an in-house far-field corpus that contains de-identified far-field data collected from millions of speakers under different conditions with users' content. This dataset contains 39 thousand hours of training data and 14 thousand hours of test data.

To support practical FL systems where only a subset of clients participate in one communication round (also known as client sampling), Self-FL can update the global model $\theta^t$ with *smoothing average*. Particularly, given an activity rate (i.e., ratio of clients selected in each round) $C \in (0, 1)$, we first obtain the current estimation $\hat{\theta}^t$ by applying Eqn. (11) to the active clients. Then, the server updates the global model using $\theta^t = (1 - C) \cdot \theta^{t-1} + C \cdot \hat{\theta}^t$. Updating the global model with a smoothing average allows us to integrate historical information and reduce instability, especially for a small activity rate.

## 4.2 Effectiveness on the image domain

**Results on MNIST and FEMNIST.** A description of MNIST and FEMNIST data are detailed in Subsection 4.1. The activity rate is set to $C = 0.1$. For MNIST, we use learning rate $\eta = 0.03$ and batch size 10 as suggested in FedProx (Li, 2020). For FEMNIST, we use $\eta = 0.01$ and batch size 10. The FL training starts from scratch and runs for 200 rounds. For comparison, we implement FedAvg and three other personalized FL techniques, DITTO (Li et al., 2021), pFedMe (Dinh et al., 2020), and PerFedAvg (Fallah et al., 2020b) with the same hyper-parameter configurations.

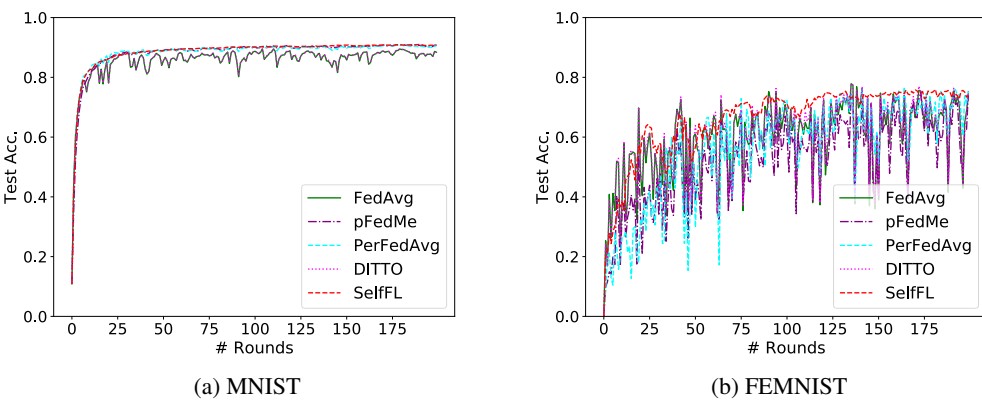

(a) MNIST

(b) FEMNIST

Figure 1: Weighted client-level test accuracy in different rounds (with $0.1$ activity rate).

For real-world data, the theoretical value of the local training steps $l_m$ computed using Eqn. (16) might be too large for the client (due to limitation of client's computation/memory budget). In this experiment, we determine the actual local training steps for Self-FL framework as $\hat{l}_m = \min\{l_m, l_{\max}\}$ where $l_{\max}$ is a pre-defined cutoff threshold. We use a fixed local training step of 20 for other FL algorithms and set $l_{\max} = 40$ for Self-FL. Note that $\sigma_m^2$ represents the variance of $\theta_m$ (i.e., model parameters such as neural coefficients). In our experiments, $\sigma_m^2$ is approximated using the empirical variance of $\theta_m$ across rounds.

Figures 1a and 1b compare the convergence performance of different FL algorithms on MNIST and FEMNIST, respectively. The horizontal axis denotes the communication round. The vertical axis denotes the weighted test accuracy of individual clients, where the weight is proportional to the sample size. We observe that: (i) Self-FL algorithm yields more *stable convergence* compared with FedAvg and the other three personalized FL methods due to our smoothing average in aggregation; (ii) Self-FL achieves the highest personalized accuracy when the global model converges.

To corroborate that the performance advantage of Self-FL does not come from a larger potential local training steps (we set $l_{max} = 40$ for Self-FL), we perform an additional experiment where we use a local step as $l = 40$ (instead of 20) for DITTO, pFedMe, and PerFedAvg on the FEMNIST dataset. In this scenario, the test accuracy of these three FL algorithms is $71.33\%$, $68.87\%$, and $71.08\%$, respectively. Our proposed method achieves a test accuracy of $76.93\%$. Furthermore, it is worth noting that other FL methods do not have a principled way to determine the value of $l$. The adaptive local steps in Self-FL also address this challenge.

**Results on CIFAR10.** We also perform experiments on the CIFAR-10 dataset (from the pFedMe repository Canh T. Dinh (2020)) with the CNN model from PyTorch example PyTorch (2022). In each round, we randomly select 5 out of 20 clients to participate in FL training. We use a learning rate of $0.01$ for all FL algorithms and set the local training step as $40$ for the baseline methods. Figure 2a visualizes the test accuracy comparison, showing that Self-FL achieves a higher accuracy.

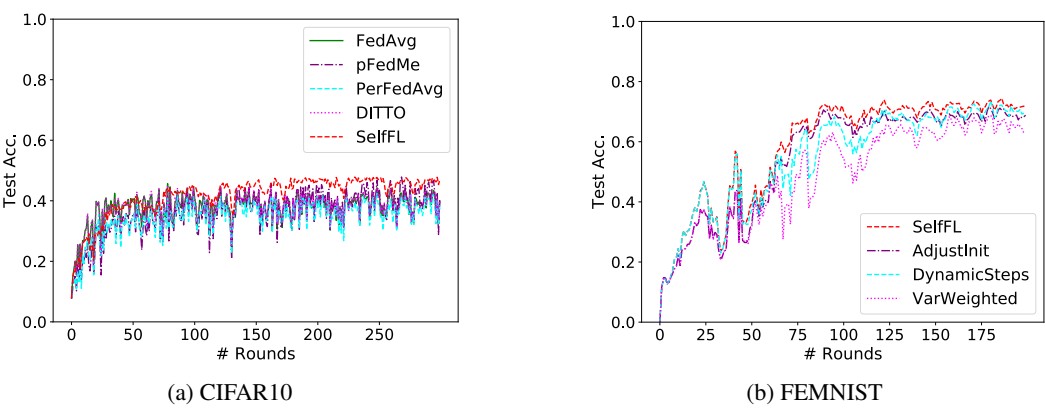

(a) CIFAR10

(b) FEMNIST

Figure 2: Weighted client-level test accuracy.

**Ablation study on Self-FL.** Recall that Self-FL consists of three key components: (i) Adjusting local initialization; (ii) Dynamic local training steps; (iii) Variance-weighted global aggregation (Section 3). To investigate the effect of each component, we perform an ablation study where we apply the full deployment of Self-FL and only only component of it. Figure 2b compares the test accuracy of the global model on FEMNIST benchmark in these four settings. We can observe that adjusting the local initialization at each round (Eqn. (10)) gives the highest contribution to Self-FL's performance.

To characterize the distribution of personalized performance across clients, we define two new metrics to evaluate different FL algorithms: (i) weighted test accuracy of clients that have top 10% most samples; (ii) worst 10% clients' average test accuracy. The evaluation results on FEMNIST (where the client sample size ranges from 1 to 206) are shown in Table 1 (standard errors in parentheses). We can observe that Self-FL outperforms other FL algorithms in terms of these auxiliary metrics. In addition, we visualize the distribution of Self-FL's personalization performance in Figure 3 by plotting the histograms of the user-level test accuracy. This user-level accuracy is obtained by evaluating the final local model (acquired by Algorithm 1) of each client on his local dataset. The same metric is measured for the baseline FedAvg. We can see from Figure 3 that the user-level test accuracy distribution under Self-FL is shifted toward the right side of its counterpart under baseline FedAvg, suggesting that Self-FL effectively improves the local model performance for individual clients and achieves the goal of personalization.

Table 1: Test accuracy comparison on FEMNIST.

| FL Alg. | FedAvg | DITTO | pFedMe | PerFedAvg | Self-FL |
|---|---|---|---|---|---|
| Top 10% most samples | 70.48% (0.07%) | 70.58% (0.14%) | 72.76% (0.28%) | 74.95% (0.56%) | 76.24% (0.07%) |
| Worst 10% user avg acc | 0% (0%) | 0% (0%) | 17.97% (1.23%) | 24.51% (1.55%) | 37.01% (1.57%) |

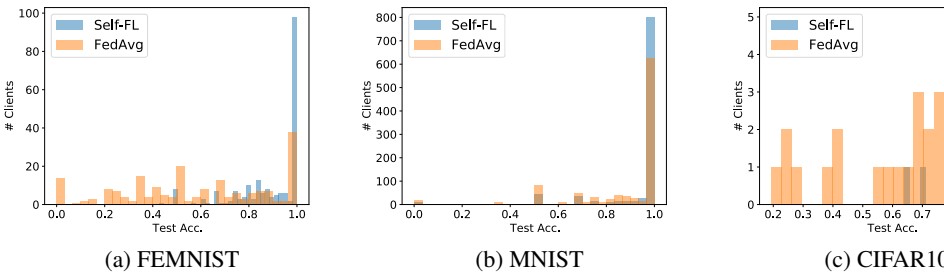

(a) FEMNIST      (b) MNIST      (c) CIFAR10

Figure 3: Distribution of the user-level personalized performance on the test set.

## 4.3 Effectiveness on the audio domain

In this section, we evaluate the performance of Self-FL on an audio dataset (Subsection 4.1) for wake-word detection. Particularly, we use a CNN that is pre-trained on the training data of different device types (i.e., heterogeneous data) as the initial global model to *warm-start* FL training for all evaluated FL algorithms. The personalization task aims to improve the wake-word detection performance at the device type level. In this experiment, we assume there are five clients in the FL system. Each client only has the training data for a specific device without data overlapping. Furthermore, all the clients participate in each communication round. Note that FedAvg (McMahan et al., 2017) and DITTO (Li et al., 2021) typically use the sample size-based weighted average to aggregate models, which do not take data heterogeneity into account. For comparison, we implement FedAvg and DITTO with both equal-weighted and sample size-based weighted model averaging (denoted by the suffix "-e' and '-w', respectively) during aggregation. For meta-learning-based PerFedAvg (Fallah et al., 2020b), we use its first-order approximation and equal-weighted aggregation. We did not report pFedMe on this dataset because we found it did not converge with various hyper-parameters.

**Evaluation metric.** One can control the trade-off between false accepts and false rejects of wake-word detection by tuning the threshold. Since we use a pre-trained model to warm-start FL, we first evaluate the detection performance of this pre-trained model as the baseline. To compare the

performance of different FL algorithms, we use the *relative false accept (FA)* value of the resulting model when the corresponding relative false reject (FR) is close to one as the metric. So a relative FA smaller than one is preferred. Here, the relative FA and FR are computed with respect to the baseline. We detail the results in two scenarios below.

Table 2a summarizes the results of the updated global model. Recall that a smaller relative FA indicates better performance. Each column reports the relative FA for a specific device type. The results show that Self-FL achieves the lowest relative FA. Several updated global models have worse performance than the baseline model, e.g., for FedAvg-w and DITTO-w in Table 2a. This is due to the setting here that local models are initialized from a pre-trained model, and the comparison is relative to that warm-start. Because clients' data are highly heterogeneous, a sample size-based aggregation rule used in FedAvg-w and Ditto-w results in a deteriorated global model compared with the original start. We provide ablation studies in the client sampling setting in the Appendix Subsection G.4, which show Self-FL still performs well.

Table 2: Detection performance (relative FA) of model on a test dataset.

| FL methods | Device Types | | | | |
|---|---|---|---|---|---|
| | A | B | C | D | E |
| Self-FL | **0.92** | **0.94** | **0.91** | **0.91** | 1.01 |
| FedAvg-w | 8.39 | 4.00 | 12.80 | 8.61 | 10.62 |
| FedAvg-e | 0.97 | 0.96 | 1.00 | 0.92 | 1.00 |
| DITTO-w | 8.38 | 4.00 | 12.75 | 8.61 | 10.23 |
| DITTO-e | 0.97 | 0.95 | 1.00 | 0.93 | **0.99** |
| PerFedAvg | 1.06 | 0.98 | 1.08 | 0.93 | 1.01 |

(a) Global model.

| FL methods | Device Type | | | | |
|---|---|---|---|---|---|
| | A | B | C | D | E |
| Self-FL | **0.93** | **0.91** | **0.90** | **0.90** | 0.99 |
| FedAvg-e | 0.95 | 0.95 | 0.93 | 0.91 | 0.98 |
| DITTO-e | 0.97 | 0.96 | 0.93 | 0.91 | 0.96 |
| PerFedAvg | 1.02 | 1.11 | 1.08 | 1.00 | **0.93** |

(b) Personalized model.

We further compare the personalization performance of Self-FL with FedAvg-e and two personalized FL algorithms, DITTO-e and PerFedAvg, on each device type's test data and summarize the results in Table 2b. We do not consider FedAvg-w and DITTO-w in this experiment since they result in performance degradation, as seen from Table 2a. One can see that Self-FL outperforms the other methods across all device types, thus demonstrating a better personalization capability.

## 5   Concluding Remarks

In this paper, we proposed Self-FL to address the challenge of balancing local model regularization and global model aggregation in personalized FL. Its key component is using uncertainty-driven local training steps and aggregation rules instead of conventional local fine-tuning and size-based weights. Overall, Self-FL connects personalized FL with hierarchical modeling and utilizes uncertainty quantification to drive personalization. Extensive empirical studies of Self-FL show its promising performance. We do not envision any negative societal impact of the work.

The **Appendix** contains further experimental studies, remarks, related works, and technical proofs.

## Acknowledgments and Disclosure of Funding

This work was done when Huili Chen worked as a Applied Scientist Intern at Alexa AI, Amazon.

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
