# A Appendix

This **Appendix** is structured as follows. We provide a summary of frequently used notations in Section B and additional related works in Section C. We show how Self-FL learns in the setting of a general parametric model in Section D. In Section E, we discuss how the uncertainty quantity in Algorithm 1 can be computed in an online fashion. In Section F, we introduce a variant of Self-FL named "Augmented Self-FL", which serves as a natural alternative to Algorithm 1. Then, in Section G, we conduct ablation studies and provide additional experimental results. Finally, we provide the proof of Proposition 3.1 in Section H.

# B Summary of Notations

We summarize the frequently used notations in Table 3.

Table 3: Frequently used notations

| Notation | Meaning |
|---|---|
| $M$ | number of clients |
| $\sigma_m^2$ | variance of client $m$'s estimated parameters, or the "intra-client uncertainty" in general |
| $\sigma_0^2$ | variance of different clients' underlying parameters, or the "inter-client uncertainty" in general |
| $\theta_m^{(\mathrm{L})}, v_m^{(\mathrm{L})}$ | posterior mean and variance of client $m$'s personal parameter |
| $\theta_m^{(\mathrm{FL})}, v_m^{(\mathrm{FL})}$ | posterior mean and variance of client $m$'s personal parameter conditional on all the data |
| $\theta^{(\mathrm{G})}, v^{(\mathrm{G})}$ | global model's posterior mean and variance |
| $\eta$ | learning rate |
| $\ell$ | learning steps |
| $t$ | FL round |
| $\theta_m^t$ | client $m$'s FL-optimal parameter |
| $\theta_m, t$ | client $m$'s local optimal parameter at round $t$ |
| $\theta^t$ | global parameter at around $t$ |

# C Further Remarks on Related Work

**Federated learning.** A general goal of FL is to train massively distributed models at a large scale (Bonawitz et al., 2019). FedAvg (McMahan et al., 2017) is perhaps the most widely adopted FL baseline, which reduces communication costs by allowing clients to train the local models for multiple iterations. To further reduce communication costs, data compression techniques such as quantization and sketching have been developed for FL (Konevcny et al., 2016; Alistarh et al., 2017; Ivkin et al., 2019). To further reduce computation costs, techniques to train a large model using small-capacity devices such as HeteroFL (Diao et al., 2020) have been developed.

**Bayesian Federated Learning.** There have been prior work that use the Bayesian perspective in FL. For instance, FedPA Al-Shedivat et al. (2020) formulates FL as a posterior inference problem where the global posterior is obtained by averaging the clients' local posteriors. FedBE Chen and Chao (2020) proposes an aggregation method by interpreting local models as samples from the global model distribution and leveraging Bayesian model ensemble to aggregate the predictions. There are two main differences between Self-FL and the above works. First, both FedPA and FedBE aim to learn a global FL model on the server side instead of personalized models for clients (the focus of our paper). Thus, the developed update rules and use scenarios are entirely different. Second, our approach is based on a novel two-level hierarchical Bayes perspective that leverages the inter-client and intra-client uncertainty to drive FL optimization. FedPA and FedBE use standard (one-level) Bayesian posterior to update the global model.

**Adaptive Federated Learning.** There has been a direction of research to automate the hyperparameter tuning process in FL. For example, FedOPT Reddi et al. (2020) proposes an FL framework with server and client optimizers to improve FL convergence instead of personalization (our focus). The adaptive server optimization in Reddi et al. (2020) is orthogonal to Self-FL. FedOPT involves tuning hyperparameters for the server's adaptive optimizer (e.g., selecting $\beta_1$, $\beta_2$, and $\epsilon$ for Adam). Also, its

aggregation rule is the same as standard FL and can be possibly replaced by our method for better performance. FedEx Khodak et al. (2021) proposes a hyperparameter tuning method inspired by weight sharing in neural architecture search. Although FedEx can auto-tune local hyperparameters, it may need extra efforts to determine its own hyperparameters, such as $\eta_t$, $\lambda_t$, and configurations $c$. In terms of the update rules, FedEx and Self-FL are also very different. In particular, a client in FedEx will sample a configuration $c$ from a hyperparameter distribution $D$, and the server will perform an exponentiated update for $D$; Self-FL will estimate empirical variances (online) to simultaneously determine local initialization, training steps, and global model aggregation.

**Heterogeneous FL.** Earlier studies of FL typically assumed that local models share the same architecture as the global model (Li et al., 2020c) to produce a single global inference model. This assumption limits the global model's complexity for the most data-indigent client. To personalize the computation and communication capabilities of each client, the work of (Diao et al., 2020) proposed a new FL framework named HeteroFL to train heterogeneous local models and still produce a single global inference model. Although this model heterogeneity can be regarded as personalization of client-side resources, it differs significantly from personalized FL in this work, where our goal is to derive client-specific models due to clients' heterogeneous data. To address system heterogeneity, methods based on asynchronous communication and active client sampling have also been developed (Bonawitz et al., 2019; Nishio and Yonetani, 2019). To encourage a fair distribution of accuracy across clients, a method based on client-weighted loss was developed in Li et al. (2019).

**Assisted learning.** Beyond edge computing, emerging real-world applications concern the collaboration among organizational learners such as research labs, government agencies, or companies. However, to avoid leaking useful and possibly proprietary information, an organization typically enforces stringent security measures, significantly limiting such collaboration. Assisted learning (AL) (Xian et al., 2020; Diao et al., 2021a;b) has been developed as a decentralized framework for organizational learners (with rich data and computation resources) to autonomously assist each other without sharing data, task labels, or models. Consequently, each learner also obtains a "personalized model" to serve its own task. A critical difference between AL and FL is that participating learners in AL do not share task labels or a global model to meet organizational model privacy Wang et al. (2021). Also, since a learner is resource-rich, AL often focuses on reducing the assistance rounds instead of communication costs at each round. As such, the techniques developed under AL are entirely different from those in the personalized FL literature.

# D   General parametric models

In a more general parametric model setting, the nature of the personalized FL problem remains the same as the simple case in Subsection 2.1. Suppose that there are $M$ clients, and their data are assumed to be generated from:

$$\theta_m \mid \theta_0 \overset{\text{IID}}{\sim} \mathcal{N}(\theta_0, \sigma_0^2), \; w_{m,i} \overset{\text{IID}}{\sim} p_m(\cdot \mid \theta_m), i \in [N_m].$$

For simplicity, we suppose that each observation $w_{m,i}$ is a scalar. The related discussions can be easily extended to multivariate settings. Compared with Eqn. (1), each client may have a different amount of data and follow client-specific distributions (denoted by $p_m$). Let $z_m \overset{\triangle}{=} \theta_m^{(\text{L})}$ denote the minimum of the negative log-likelihood function $L_m(\theta_m) = \sum_{i=1}^{N_m} \text{loss}(w_{m,i}, \theta_m)$. Standard asymptotic statistics for $M$-estimators (Van der Vaart, 2000, Ch.5) under regularity conditions show that $\sqrt{N_m}(z_m - \theta_m) \to_d \mathcal{N}(0, v_m^2)$ as $N_m \to \infty$, with a constant $v_m^2$ (the inverse Fisher information). Thus, we approximately have:

$$z_m \mid \theta_m \sim \mathcal{N}(\theta_m, N_m^{-1} v_m^2).$$

In other words, we may treat the statistic $\theta_m^{(L)}$ as the "data" $z_m$ in the two-level Gaussian model in Subsections 2.1. Letting $\sigma_m^2 \overset{\triangle}{=} N_m^{-1} v_m^2$ and taking it into the equations in Subsection 2.1, we can approximate posterior quantities accordingly. Also, the objective function $L_m(\theta_m)$ is approximated by its second-order Taylor expansion in the form of Eqn. (4). It is worth mentioning that the asymptotic results for parametric models may not hold for neural networks.

In particular, we let:

$$\theta_1^{(\text{FL})} \triangleq \frac{N_1 v_1^{-2} \theta_1^{(\text{L})} + \sum_{m \neq 1} (\sigma_0^2 + N_m^{-1} v_m^2)^{-1} \theta_m^{(\text{L})}}{N_1 v_1^{-2} + \sum_{m \neq 1} (\sigma_0^2 + N_m^{-1} v_m^2)^{-1}},$$

$$\theta^{(\text{G})} \triangleq \frac{\sum_{m \in [M]} (\sigma_0^2 + N_m^{-1} v_m^2)^{-1} \theta_m^{(\text{L})}}{\sum_{m \in [M]} (\sigma_0^2 + N_m^{-1} v_m^2)^{-1}}.$$

Each client, say client 1, can obtain its personal optimal solution $\theta_1^{(\text{FL})}$ through the following initial value and optimization parameters $\eta, \ell$.

$$\theta_1^{\text{INIT}} = \frac{\sum_{m \neq 1} (\sigma_0^2 + N_m^{-1} v_m^2)^{-1} \theta_m^{(\text{L})}}{\sum_{m \neq 1} (\sigma_0^2 + N_m^{-1} v_m^2)^{-1}}, \tag{15}$$

$$(1 - N_1 v_1^{-2} \eta)^\ell = \frac{\sum_{m \neq 1} (\sigma_0^2 + N_m^{-1} v_m^2)^{-1}}{N_1 v_1^{-2} + \sum_{m \neq 1} (\sigma_0^2 + N_m^{-1} v_m^2)^{-1}}. \tag{16}$$

*Remark* D.1 (**Interpretations of the choice of $\eta$ and $\ell$**). Let us consider the following extreme cases.

● (Large sample size, few clients) Suppose that $N_1 = \cdots = N_M \gg M > 1$, and $v_1^2, \sigma_0^2$ are comparable. Then, for client 1, Eqn. (16) becomes:

$$(1 - N_1 v_1^{-2} \eta)^\ell \approx \frac{(M-1)\sigma_0^{-2}}{N_1 v_1^{-2} + (M-1)\sigma_0^{-2}} \approx \frac{M}{N_1} \frac{v_1^2}{\sigma_0^2},$$

which implies that client 1 needs to run aggressively with its local data. Also, the smaller $\sigma_1^2 = v_1^2/N_1$ (namely better data quality or larger sample size), the more local training steps is favored.

● (Small sample size, many clients) Suppose that $1 \ll N_1 = \cdots = N_M \ll M$, and $v_1^2 = \cdots = v_M^2$. We have:

$$(1 - N_1 v_1^{-2} \eta)^\ell \approx \frac{(M-1)(\sigma_0^2 + N_1^{-1} v_1^2)^{-1}}{N_1 v_1^{-2} + (M-1)(\sigma_0^2 + N_1^{-1} v_1^2)^{-1}}$$

$$\approx 1 - \frac{N_1}{M} \frac{\sigma_0^2}{v_1^2}, \tag{17}$$

which implies that client 1 needs to set:

$$N_1 v_1^{-2} \eta \cdot \ell \approx \frac{N_1}{M} \frac{\sigma_0^2}{v_1^2}, \quad \text{or} \quad \eta \cdot \ell \approx \frac{\sigma_0^2}{M}.$$

It is interesting to see that the choice of $\eta, \ell$ is independent of the client in this scenario.

● (Homogeneous clients) Suppose that $\sigma_0^2 = 0$, meaning no discrepancy between clients' data distributions. We have

$$(1 - N_1 v_1^{-2} \eta)^\ell = \frac{\sum_{m \neq 1} N_m v_m^{-2}}{\sum_{m \in [M]} N_m v_m^{-2}}.$$

If we further assume that $v_1 = \cdots = v_M$, $M$ is large, and $N_1, \ldots, N_M$ are at the same order, client 1 needs to set $N_1 v_1^{-2} \eta \cdot \ell \approx N_1/N$, where $N \triangleq N_1 + \cdots + N_M$. In other words, $\eta \cdot \ell \approx v_1^2/N$, which is the same among all clients.

# E   Implementation Details of Algorithm 1

In Algorithm 1, we need to evaluate the empirical variance of a set of models. This can be a memory concern for edge devices in practice. We use the following way to perform online calculation, so that the required hardware memory does not grow with the number of rounds or clients.

To calculate the sample variance

$$\widehat{\sigma_t^2} \triangleq \frac{1}{t} \sum_{i \in [t]} (x_i - \bar{x}_t)^2 \tag{18}$$

where $\bar{x}_t \triangleq t^{-1} \sum_{i \in [t]} x_i$, we do not need to store $x_1, \ldots, x_t$. Instead, we only need to store $\bar{x}_t$ at time step $t$, and calculate the sample variance in the following recursive way.

$$
\begin{aligned}
&\text{For } t = 1, 2, \ldots \\
&\bar{x}_t = \frac{t-1}{t} \bar{x}_{t-1} + \frac{1}{t} x_t, \\
&\widehat{\sigma_t^2} = \frac{t-1}{t} \widehat{\sigma_{t-1}^2} + \frac{t-1}{t} (\bar{x}_t - \bar{x}_{t-1}) + \frac{(x_t - \bar{x}_t)^2}{t},
\end{aligned} \tag{19}
$$

with $\bar{x}_0 = \widehat{\sigma_0^2} = 0$. It can be verified that the $\widehat{\sigma_t^2}$ in (19) is equivalent to that in (18). The recursive computation is particularly favorable for large neural networks with millions of parameters and small hardware memory.

*Remark* E.1 (Client sampling). Suppose that in each round, only a subset of clients is activated. The subscript $t$ in the online update of $\sigma_m^2$ is client $m$'s the local time counter, namely the total counts that client $m$ is activated in FL communications. This local time counter shall be distinguished from the global counter $t$ that indicates the system communication round. Also, a client's intra-client uncertainty is only online updated in those rounds that it is activated.

## F    An Alternative Version of Algorithm 1

In this section, we present a slightly more complicated version of Self-FL in Algorithm 2, as an alternative of Algorithm 1. Its pseudocode can be found at the end of the Appendix, and the main changes are highlighted with blue fonts. The motivation goes back to the discussions in Subsection 3.2, where we mentioned the use of a client $m$'s local parameter at each round as a bootstrapped estimate of the underlying $\theta_m^{(L)}$. In Algorithm 2, each client needs to run additional local training until convergence (denoted by $\theta_{m,t}$) in each FL round and communicates two local models with the server.

This alternative variant requires the computation and communication of local-optimal solutions ($\theta_{m,t}$). Thus, the implementation of Self-FL in Algorithm 2 incurs higher computation costs for local clients (due to additional local fine-tuning) and doubles the communication overhead compared with the standard FedAvg (McMahan et al., 2017). We will perform ablation studies in Subsection G.2 to compare these two algorithms.

## G    Additional Experiments

We perform extended experiments of Self-FL and summarize the results in this section. Particularly, in Subsection G.2, we compare the performance of two implementation variants of Self-FL. In Subsection G.4, we revisit the audio data in a client sampling setting, as a supplement to the full participation setting studied in Subsection 4.3. In Subsections G.5 and G.6, we show the impact of the client activity rate $C \in (0, 1]$ and provide insights into the adaptive local training steps of Self-FL, respectively.

### G.1    Convergence study on synthetic data

For FedAvg on the synthetic dataset, we use a gradient descent-based local update rule: $\theta_1^l = \theta_1^{l-1} - \eta(\theta_1^{l-1} - \theta_1^{(L)})/\sigma_1^2$. For DITTO, an additional regularization term is used and the update rule is: $\theta_1^l = \theta_1^{l-1} - \eta(\theta_1^{l-1} - \theta_1^{(L)})/\sigma_1^2 - \lambda(\theta_1^{l-1} - \theta^{(G)})$. A skeptical reader may wonder whether DITTO is equivalent to Self-FL for a particular choice of $\lambda$. This is not the case, even in our earlier example of the two-layer Gaussian model. The two methods differ in the aggregation rule and the initialization of each client.

**Case 1: Homogeneous setting.** We use a small inter-client uncertainty in this setup and assume that clients have similar local sample sizes. There are $M = 20$ clients in the FL system where the local sample size $N_m$ is uniformly sampled from the range $[10, 20]$. We set the parent parameter $\theta_0 = 1.6$. The inter-client and intra-client uncertainty are $\sigma_0^2 = 0.001$, $\sigma_m^2 = 0.1$, respectively. The experiment is repeated 200 times, and we report the mean value as the final metric. The standard error of the experiment is within 0.001. Note that the effective $\sigma_m^2$ shall be divided by $N_m$.

Figure 4 compares the parameter estimation performance of Self-FL algorithm with FedAvg and three existing personalized FL techniques: DITTO (Li et al., 2021), pFedMe (Dinh et al., 2020), and

**Algorithm 2** Self-Aware Personal Federated Learning by Introducing Local Minimum $\theta_{m,t}$ (Augmented Self-FL).
The main differences with Algorithm 1 are highlighted with blue fonts.

---

**Input:** A server and $M$ clients. System-wide objects, including the initial model parameters $\theta_m^0 = \theta_{m,0} = \theta^0$, initial uncertainty of clients' parameters $\widehat{\sigma_0^2} = 0$, the activity rate $C$, the number of communication rounds $T$, server batch size $B_s$, and client batch size $B_m$, and the loss function $(z, \theta) \mapsto \text{loss}(z, \theta)$ corresponding to a common model architecture. Each client $m$'s local resources, including $N_m$ data observations, parameter $\theta_m$, number of local training steps $\ell_m$, and learning rate $\eta_m$.

**System executes:**
  **for** each communication round $t = 1, \ldots T$ **do**
    $\mathbb{M}_t \leftarrow \max(\lfloor C \cdot M \rfloor, 1)$ active clients uniformly sampled without replacement
    **for** each client $m \in \mathbb{M}_t$ **in parallel do**
      Distribute server model parameters $\theta^{t-1}$ to local client $m$
      $(\theta_m^t, \theta_{m,t}, \widehat{\sigma_m^2}) \leftarrow \textbf{ClientUpdate}(z_{m,1}, \ldots, z_{m,N_m}, \theta^{t-1}, \widehat{\sigma_0^2})$
    $(\theta^t, \widehat{\sigma_0^2}) \leftarrow \textbf{ServerUpdate}(\theta_m^t, \theta_{m,t}, \widehat{\sigma_m^2}, m \in \mathbb{M}_t)$

**ClientUpdate** $(z_{m,1}, \ldots, z_{m,N_m}, \theta^{t-1}, \widehat{\sigma_0^2})$**:**
  Calculate the uncertainty of the local optimal solution $\theta_m^{(\text{L})}$ of client $m$:

$$\widehat{\sigma_m^2} = \text{empirical variance of } \{\theta_{m,1}, \ldots, \theta_{m,t-1}\}. \tag{20}$$

  Calculate the initial parameter

$$\theta_m^t \triangleq \theta^{t-1} - \frac{(\widehat{\sigma_0^2} + \widehat{\sigma_m^2})^{-1}}{\sum_{k:k \neq m}(\widehat{\sigma_0^2} + \widehat{\sigma_k^2})^{-1}}(\theta_{m,t-1} - \theta^{t-1}), \tag{21}$$

  Choose $\ell_m \geq 1$ according to

$$\left(1 - \frac{\eta_m}{B_m \widehat{\sigma_m^2}}\right)^{\ell_m} = \frac{\sum_{k:k \neq m}(\widehat{\sigma_0^2} + \widehat{\sigma_k^2})^{-1}}{1/\widehat{\sigma_m^2} + \sum_{k:k \neq m}(\widehat{\sigma_0^2} + \widehat{\sigma_k^2})^{-1}}$$

  **for** local step $s$ from 1 to $\ell_m$ **do**
    Sample a batch $I \subset [N_m]$ of size $B_m$
    Update $\theta_m^t \leftarrow \theta_m^t - \eta_m \nabla_\theta \sum_{i \in I} \text{loss}(w_{m,i}, \theta_m^t),$
  Let $\theta_{m,t} \leftarrow \theta_m^t$.
  **for** local step $s$ from $\ell_m + 1$ to $\infty$ **do**
    Sample a batch $I_s \subset [N_m]$ of size $B_m$
    Update $\theta_{m,t} \leftarrow \theta_{m,t} - \eta_m \nabla_\theta \sum_{i \in I_s} \text{loss}(w_{m,i}, \theta_{m,t}),$
    If it converges, then Break.
  Store $\theta_{m,t}$ locally
  Return $(\theta_m^t, \theta_{m,t}, \widehat{\sigma_m^2})$ and send them to the server

**ServerUpdate** $(\theta_m^t, \theta_{m,t}, \widehat{\sigma_m^2}, m \in \mathbb{M}_t)$**:**
  Calculate the uncertainty of clients' underlying parameters

$$\widehat{\sigma_0^2} = \text{empirical variance of } \{\theta_{m,t}, m \in \mathbb{M}_t\}. \tag{22}$$

  Calculate model parameters

$$\theta^t \triangleq \frac{\sum_{m \in \mathbb{M}_t}(\widehat{\sigma_0^2} + \widehat{\sigma_m^2})^{-1}\theta_{m,t}}{\sum_{m \in \mathbb{M}_t}(\widehat{\sigma_0^2} + \widehat{\sigma_m^2})^{-1}}. \tag{23}$$

  Return $(\theta^t, \widehat{\sigma_0^2})$

---

PerFedAvg (Fallah et al., 2020b). The estimation error is measured in $\ell_1$ norm. All FL algorithms use the same learning rate $10^{-4}$ and are trained for 200 rounds. For Self-FL, the local training steps $l_m$'s are computed using Eqn. (16). For other FL methods, the local training steps are set to 50.

We consider the activity rate $C \in \{0.1, 0.2, 0.5, 0.8, 1\}$ and compare different algorithms in each setting. We draw two conclusions from Figure 4. (i) When there are sufficient active clients in each communication round, Self-FL and PerFedAvg achieve the lowest estimation error for both global and local parameters compared with others. (ii) When the number of active clients is small, Self-FL outperforms the other methods.

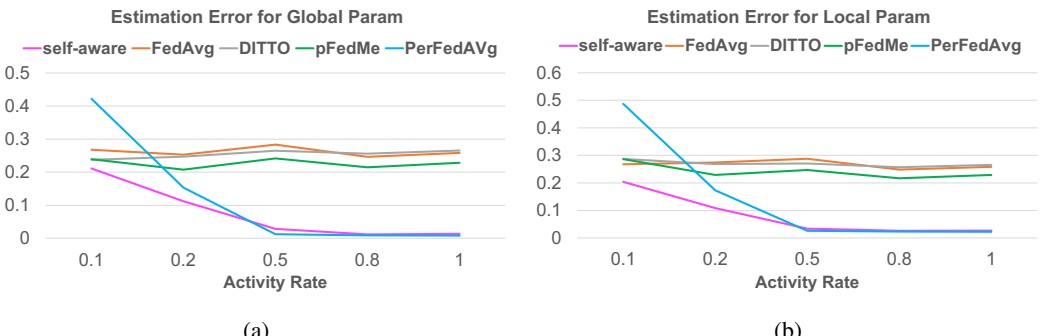

(a)  (b)

Figure 4: Performance of different FL algorithms on synthetic data of high homogeneity. Estimations errors of the global and local parameters (averaged over clients) are shown in (a) and (b), respectively.

**Case 2: Heterogeneous setting.** To generate heterogeneous data across clients, we use a large value for inter-client uncertainty $\sigma_0^2$ and assume that clients have different scales of local samples size $N_m$. We consider $\sigma_0^2 = 1$ and $\sigma_m^2 = 0.1$. The sample size of each client is uniformly drawn from $[10, 200]$ (which is much broader than that in Case 1). Other settings are the same as Case 1.

Recall that if the inter-client uncertainty is larger, the clients' data are more irrelevant, and the FL posterior approaches the local posterior. Then, the client learns on its own. The global model becomes a simple equal-weighted averaging of local models (detailed in Remark 2.1). We show the performance comparison results of different FL algorithms on the heterogeneous dataset in Figure 5. One can see from the figure that Self-FL's estimation of both the global parameter and the local ones are insensitive to the client participation ratio.

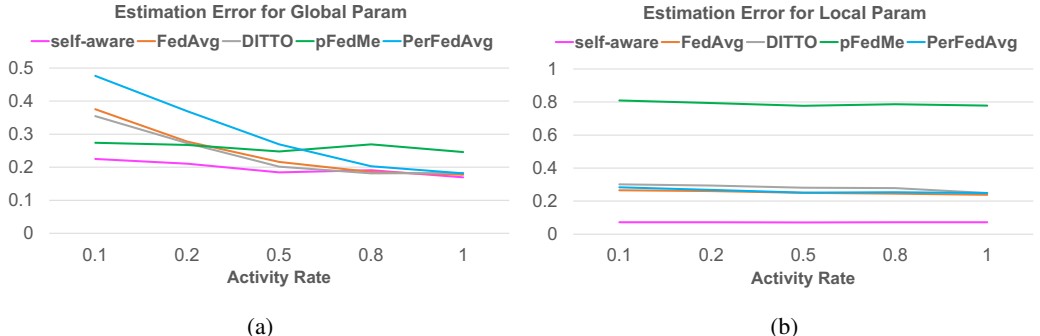

(a)  (b)

Figure 5: Performance comparison on synthetic datasets of high heterogeneity. Estimation errors are shown for global parameter (a) and local parameter (b).

### G.2 Ablation study of Self-FL's performance with Algorithms 1 and 2

**Results on synthetic data.** In this experiment, we conduct an ablation study of Self-FL's performance with two different implementation variants outlined in Algorithm 1 (using the early-stop parameter $\theta_m^t$) and Algorithm 2 (using the sufficiently trained local parameter $\theta_{m,t}$). We use the same synthetic datasets and learning parameters as Section 4. We repeat the experiments for 200 times and report the mean value. The standard error is kept within 0.001. The results are visualized in Figure 6. One can see from the comparison that the performance gap between these two variants is negligible on both homogeneous and heterogeneous data.

**Results on MNIST and FEMNIST images.** Figure 7 shows the evaluation results, where the *y*-axis denotes the weighted training accuracy of all personalized models. The two curves in Figure 7b are very close and the accuracy difference is small than 1%.

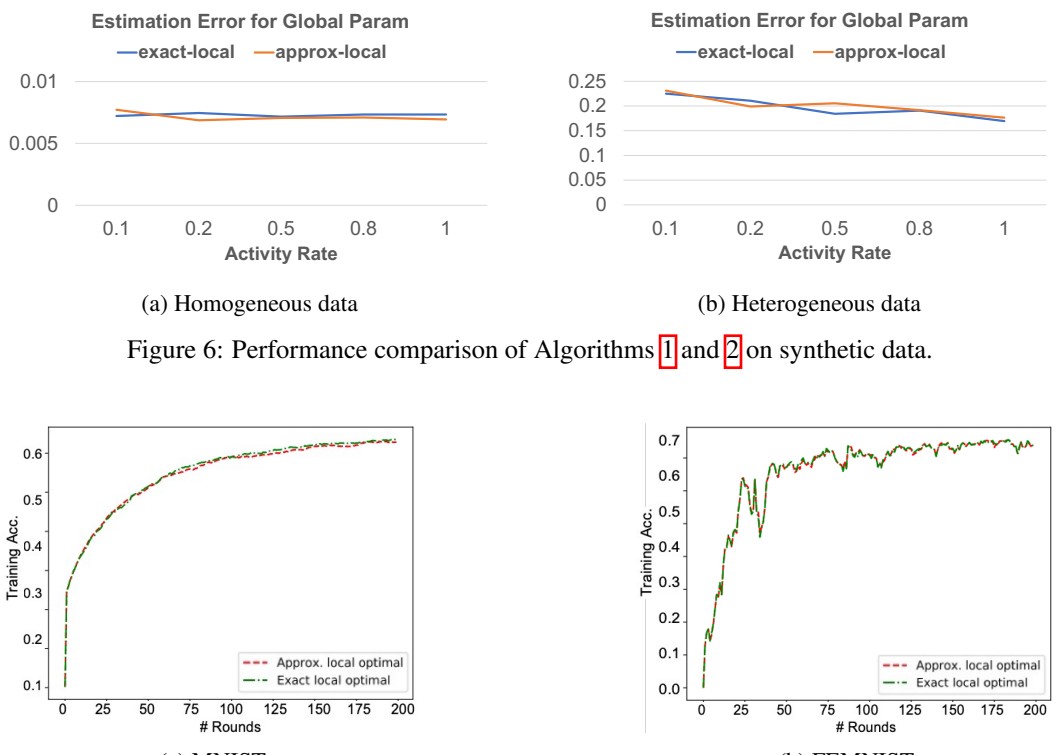

(a) Homogeneous data                    (b) Heterogeneous data

Figure 6: Performance comparison of Algorithms 1 and 2 on synthetic data.

(a) MNIST                    (b) FEMNIST

Figure 7: Performance comparison of Algorithms 1 and 2 on the MNIST and FEMNIST image data.

**Results on the Amazon Alexa audio dataset.** We also perform ablation experiments to assess the performance of two versions of Self-FL on the wake-word audio dataset. The results in Table 4 show that their performance is similar.

Table 4: Detection performance (relative FA) of the *global model* on the test dataset of wake-word audio data. The devices are in the normal state.

| FL methods | Device Types | | | | |
|---|---|---|---|---|---|
| | **A** | **B** | **C** | **D** | **E** |
| **Self-FL (Algorithm 2)** | 0.93 | 0.94 | 0.96 | 0.89 | 0.98 |
| **Self-FL (Algorithm 1)** | 0.92 | 0.94 | 0.91 | 0.91 | 1.01 |

### G.3 Experiments on Sent140 Dataset

Sent140 Go et al. (2009) is a text sentiment classification dataset. We generate non-i.i.d. FL training data of Sent140 following the same procedure as FedProx Li (2020). As for the ML model, we use a two-layer LSTM binary classifier for this experiment. The model has two LSTM layers with 256 hidden units, and the last layer is a fully-connected layer. We use the pre-trained 300D GloVe embedding Pennington et al. (2014) to transform the input sequence of 25 characters into the embedding space. For each FL algorithm, we train the local models with a learning rate of $0.3$, a batch size of 10, and randomly sub-sample 77 clients in each communication round (corresponding to a client activity rate $C = 0.1$). Each selected local client trains his model for 2 epochs. This hyper-parameter configuration is suggested in the previous paper Li et al. (2020b). Since the number of local epochs is small, we skip Self-FL's computation of the local training steps $l_m$ and use the same local epochs $l_m = 2$.

We use *warm start* in the Sent140 experiment. Particularly, we train a global model from scratch with FedAvg for 200 rounds and use it as the initialization of the global model for all the other FL algorithms. With this warm start, we continue training the global model with various FL algorithms for another 400 rounds. Figure 8 shows the training and test accuracy of the local model obtained by

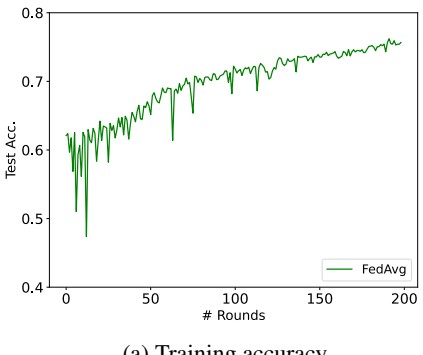
(a) Training accuracy.

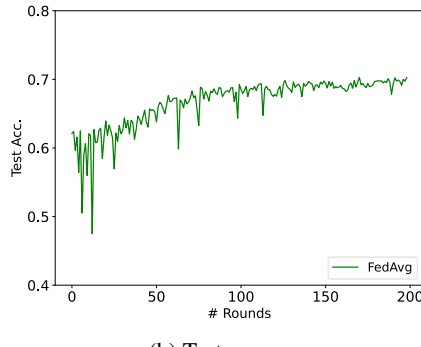
(b) Test accuracy.

Figure 8: Performance of FL training from scratch using FedAvg (*warm-start* stage).

FedAvg McMahan et al. (2017). We report the weighted accuracy across all users in the FL system where the weight ratio of each user is proportional to his local sample size.

With the pre-trained global model from FedAvg (at round 200) as the 'warm-start' initialization, we continue FL training with different algorithms for another 400 rounds. Figure 9 compares the training and test accuracy of the personalized/local models obtained by different FL algorithms. The accuracy is aggregated across clients where the aggregation weight is proportional to the local sample size. Note that Figure 9 shows the accuracy change from round 200 (where warm start ends) to round 600. We can see from Figure 9a that both Self-FL and FedAvg demonstrates better convergence performance compared to DITTO Li et al. (2021), pFedMe Dinh et al. (2020), and PerFedAvg Fallah et al. (2020b). Recall that the first 200 rounds are trained with FedAvg and continued by other FL algorithms. As the result, the test accuracy in Figure 9b does not change much from round 200 to round 600 since it 'saturates' during the warm-start stage (Figure 8b).

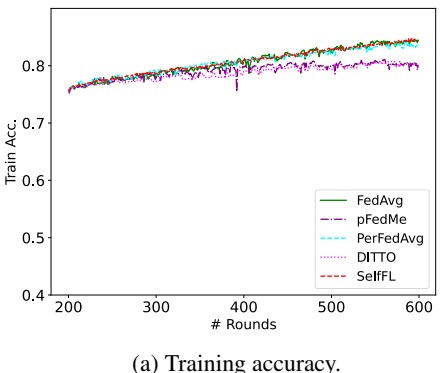
(a) Training accuracy.

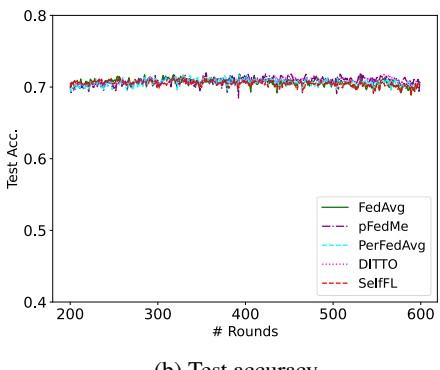
(b) Test accuracy.

Figure 9: Performance comparison of different FL algorithms on Sent140 text data. Note that the x-axis starts from round 200 since we use warm-start in this experiment.

### G.4 Self-FL's performance on wake-word data in clients selection setting

In this experiment, we evaluate the performance of FL algorithms in a client selection scenario. More specifically, we assume there are 4 clients for each of the five device types (A~E). Each client has a uniform partition of the training set for a specific device type. The FL system has a total of 20 clients. We set the activity rate to $C = 0.25$.

We consider two variants of Self-FL in the clients selection scenario: (i) Client-level variant. This implementation aggregates local models from the currently active clients in each round. (ii) Cluster-level variant. This method groups clients into clusters and then perform aggregation at the cluster level. In our experiments, we cluster clients using device types. We can also use the intra-client uncertainty ($\sigma_m^2$) as the clustering criteria since this information is available for the server (Algorithm 1). With the technique of clients clustering, the server stores the latest model parameter for each cluster as its 'representative'. In each communication round, the server loads the weight for each non-active cluster.

The global model is updated by aggregating across all clusters. It is worth noting that storing the latest model for each cluster is much more scalable than saving the model separately for each client.

Table 5: Detection performance (relative FA) of the global model obtained using two variants of Self-FL on test data.

| FL methods | State | Device Type | | | | |
| --- | --- | --- | --- | --- | --- | --- |
| | | A | B | C | D | E |
| Self-FL (client-level) | normal | 0.91 | 0.92 | 0.97 | 0.91 | 0.99 |
| | playback | 0.93 | 0.96 | 1.80 | 1.00 | 1.00 |
| | alarm | NA | 0.89 | NA | 1.04 | 1.00 |
| Self-FL (cluster-level) | normal | 0.96 | 0.95 | 1.00 | 0.91 | 0.97 |
| | playback | 0.86 | 0.99 | 1.00 | 0.97 | 1.01 |
| | alarm | NA | 0.89 | NA | 1.00 | 1.00 |

The performance of the above two variants is shown in Table 5. We can see that the cluster-level aggregation obtains more conservative global models, and its performance improvement is more stable across different device types and operation states compared to the client-level variant. This observation is intuitive, because keeping track of the latest weights for each cluster representative for aggregation can prevent the global model from catastrophic forgetting when only a small subset of clients are active in each round.

## G.5 Impact of the activity rate $C$

We use the client activity rate as the weight for smoothing the average in the above experiments. While this yields a more stable update, it could cause a slow convergence. To investigate the impact of the weights in our smoothing average update for the global model, we set the activity rate to $0.02$ and compare the performance of Self-FL with FedAvg and DITTO. Figure 10 shows the empirical results in this sparse client participation case. One can see that Self-FL converges slower than FedAvg and DITTO when we use the activity rate (0.02) as the weight for the current estimation of the global model in smoothing average. This is expected since a small weight for $\hat{\theta}^t$ means that the update of the global model is more conservative.

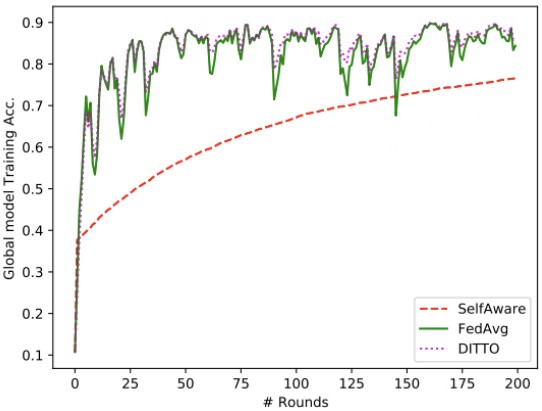

Figure 10: Convergence performance of different FL algorithms on the MNIST dataset when the activity rate is set to $0.02$.

We also perform an ablation study on the wake-word dataset to investigate the impact of the activity rate. The results are summarized in Table 6. We use the client-level implementation of Self-FL in Subsection G.4 in this experiments. one can see that Self-FL framework obtains better wake-word detection performance (i.e., smaller relative FA values) when the activity rate is not too small.

Table 6: Self-FL's performance with two different activity rates on the wake-word dataset. The relative FA values of the updated global models are reported.

| Activity Rate | State | Device Types | | | | |
| --- | --- | --- | --- | --- | --- | --- |
| | | A | B | C | D | E |
| | normal | **0.91** | 0.92 | 0.97 | 0.91 | 1.00 |
| C = 0.25 | playback | 0.93 | 0.96 | 1.80 | 1.00 | 1.00 |
| | alarm | NA | 0.89 | NA | 1.04 | 1.00 |
| | normal | 0.95 | 0.95 | 1.00 | 0.91 | 0.97 |
| C = 0.1 | playback | 0.86 | 0.97 | 1.00 | 0.96 | 1.01 |
| | alarm | NA | 0.89 | NA | 1.00 | 1.00 |

## G.6 Study of Self-FL's adaptive local training

A key component of Self-FL is the data-adaptive local training procedure as discussed in Section 3. The local optimization trajectory is designed in such a way that it achieves a good balance between local model improvement and global model update. We study the theoretical values of local training steps $l_m$ computed using Eqn. (8). Figure 11 shows the histogram of $l_m$ of the active clients in round $t = 100$. We also visualize the dynamics of the local training steps for a randomly selected client across FL rounds in Figure 12. The value of $l_m$ exists for a subset of rounds since we use the activity rate $C = 0.1$. We can see that the computed number of steps $l_m$ has an overall decreasing trend after round 150, suggesting a drop of the intra-client uncertainty $\sigma_m^2$. In other words, the quality of personalized model $\theta_m^t$ improves eventually.

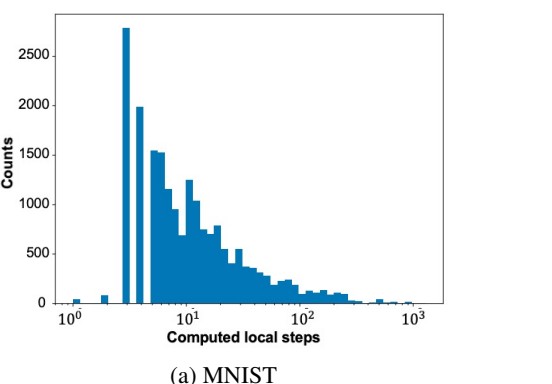
(a) MNIST

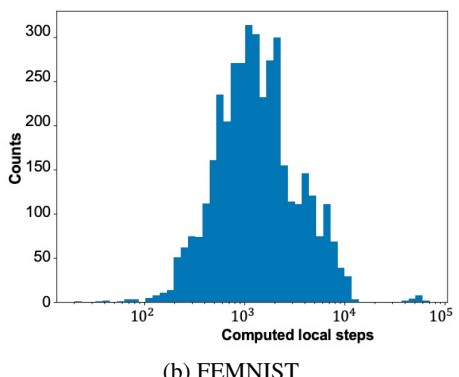
(b) FEMNIST

Figure 11: Histogram of the computed number of local training steps for active clients at the round $t = 100$.

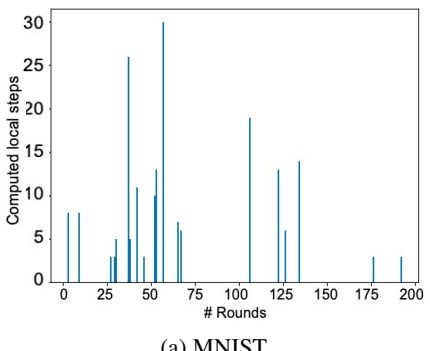
(a) MNIST

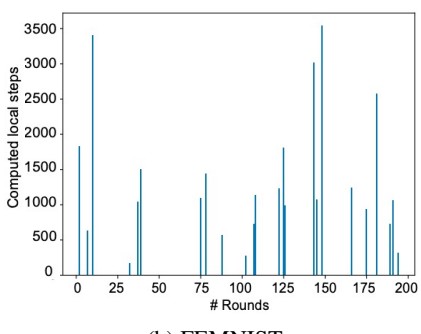
(b) FEMNIST

Figure 12: Trajectory of the computed local training steps for a randomly selected client in different FL rounds.

## G.7  Limitations of Self-FL and Future Directions

There are some new problems left from the work that deserve further research. First, in many practical applications, clients only have unlabeled data, and it is not easy to annotate those data. This poses a significant challenge in training personalized models and evaluating them. An important problem is to integrate self-supervised FL techniques Diao et al. (2021c) into personalization to address such a lack-of-label issue. Second, we considered heterogeneity in terms of clients' data distributions. There are other aspects of heterogeneity, e.g., those at the system and model levels. A more integrated view of FL heterogeneity is lacking. Third, a critical problem in personalized FL is to assess whether any particular client model, for a given set of local data, has achieved its theoretical limit from, e.g., a goodness-of-fit perspective Zhang et al. (2021). We refer to Ding et al. (2022) for an outlook on other challenges related to personalized FL.

# H  Proof of Proposition 3.1

In this section, we provide detailed proof of Proposition 3.1 introduced in the main paper. For brevity, we write $\sum_{k \in [M]}$ and $\sum_{k \in [M] - \{m\}}$ as $\sum_k$ and $\sum_{k: k \neq m}$, respectively. We define $\sum_{i,j: i \neq j}$ similarly.

Let $w_m = (\sigma_m^2 + \sigma_0^2)^{-1}$. According to the assumption, let $c_v$ be an upper bound of $\max_{m \in [M]} \sigma_m^2$. Then we have

$$w_{\min} \triangleq (c_v + \sigma_0^2)^{-1} \leq \min_{m \in [M]} w_m \leq \max_{m \in [M]} w_m \leq \sigma_0^{-2} \triangleq w_{\max}. \tag{24}$$

Let $v = \max_{m \in [M]} (w_m / \sum_k w_k)$. It follows from (24) that

$$v = O(M^{-1}) \tag{25}$$

for large $M$. Let

$$\zeta_m^t \triangleq \hat{\theta}_m^t - \theta_m^{(\text{FL})}, \quad \forall m \in [M], t = 0, 1, \ldots \tag{26}$$

Recall that by the assumption, $C$ is a constant upper bound of $\max_{m \in [M]} |\zeta_m^0|$. Next, we will provide a uniform bound on $|\zeta_m^t|$, by considering every $t \geq 1$.

According to the update rule, as shown in equation (6), we have

$$\theta_m^t = \frac{\sigma_m^{-2}}{\sigma_m^{-2} + \sum_{k: k \neq m} w_k} \theta_m^{(\text{L})} + \frac{\sum_{k: k \neq m} w_k}{\sigma_m^{-2} + \sum_{k: k \neq m} w_k} \theta_m^{t, \text{INIT}}, \tag{27}$$

where $\theta_m^{t, \text{INIT}}$ is the initial parameter of client $m$ at time $t$, introduced in (10). Recall from (3) that

$$\theta_m^{(\text{FL})} = \frac{\sigma_m^{-2} \theta_m^{(\text{L})} + \sum_{k: k \neq m} w_k \theta_k^{(\text{L})}}{\sigma_m^{-2} + \sum_{k: k \neq m} w_k}. \tag{28}$$

Combining (27) and (28), and invoking the definition in (12), we have

$$|\zeta_m^t| = |\theta_m^t - \theta_m^{(\text{FL})}| = \left| \frac{\sum_{k: k \neq m} w_k}{\sigma_m^{-2} + \sum_{k: k \neq m} w_k} \left( \theta_m^{t, \text{INIT}} - \frac{\sum_{k: k \neq m} w_k \theta_k^{(\text{L})}}{\sum_{k: k \neq m} w_k} \right) \right| \tag{29}$$

$$\leq q \left| \theta_m^{t, \text{INIT}} - \frac{\sum_{k: k \neq m} w_k \theta_k^{(\text{L})}}{\sum_{k: k \neq m} w_k} \right| \tag{30}$$

By the definition of $\theta_m^{t, \text{INIT}}$, we may rewrite it as

$$\theta_m^{t, \text{INIT}} = \theta^{t-1} - \frac{(\sigma_0^2 + \sigma_m^2)^{-1}}{\sum_{k: k \neq m} (\sigma_0^2 + \sigma_k^2)^{-1}} (\theta_m^{t-1} - \theta^{t-1}) = \frac{\sum_{k: k \neq m} w_k \theta_m^{t-1}}{\sum_{k: k \neq m} w_k}. \tag{31}$$

Therefore, with the definition in (26), we have

$$\theta_m^{t,\text{INIT}} - \frac{\sum_{k:\,k\neq m} w_k \theta_k^{(\text{L})}}{\sum_{k:\,k\neq m} w_k} = \frac{\sum_{k:\,k\neq m} w_k(\theta_k^{t-1} - \theta_k^{(\text{L})})}{\sum_{k:\,k\neq m} w_k} \tag{32}$$

$$= \frac{\sum_{k:\,k\neq m} w_k(\zeta_k^{t-1} + \theta_k^{(\text{FL})} - \theta_k^{(\text{L})})}{\sum_{k:\,k\neq m} w_k} \tag{33}$$

$$= \frac{\sum_{k:\,k\neq m} w_k \zeta_k^{t-1}}{\sum_{k:\,k\neq m} w_k} + \frac{\sum_{k:\,k\neq m} w_k \theta_k^{(\text{FL})}}{\sum_{k:\,k\neq m} w_k} - \frac{\sum_{k:\,k\neq m} w_k \theta_k^{(\text{L})}}{\sum_{k:\,k\neq m} w_k}. \tag{34}$$

Meanwhile, it can be verified that

$$\frac{\sum_{k:\,k\neq m} w_k \theta_k^{(\text{FL})}}{\sum_{k:\,k\neq m} w_k} = \frac{\sum_k w_k \theta_k^{(\text{FL})}}{\sum_k w_k}(1 - v_m)^{-1} - \frac{v_m}{1 - v_m}\theta_m^{(\text{FL})}, \quad \text{where } v_m \triangleq \frac{w_m}{\sum_k w_k} \leq v, \tag{35}$$

$$\frac{\sum_{k:\,k\neq m} w_k \theta_k^{(\text{L})}}{\sum_{k:\,k\neq m} w_k} = \frac{\sum_k w_k \theta_k^{(\text{L})}}{\sum_k w_k}(1 - v_m)^{-1} - \frac{v_m}{1 - v_m}\theta_m^{(\text{L})}, \tag{36}$$

which further implies that

$$\left| \frac{\sum_{k:\,k\neq m} w_k \theta_k^{(\text{FL})}}{\sum_{k:\,k\neq m} w_k} - \frac{\sum_{k:\,k\neq m} w_k \theta_k^{(\text{L})}}{\sum_{k:\,k\neq m} w_k} \right| \leq \frac{1}{1-v}|D| + \frac{2vc_\theta}{1-v}, \tag{37}$$

$$\text{where } D \triangleq \frac{\sum_k w_k(\theta_k^{(\text{FL})} - \theta_k^{(\text{L})})}{\sum_k w_k}, \tag{38}$$

and $c_\theta$ is an upper bound of $|\theta_m^{(\text{FL})}|$ and $|\theta_m^{(\text{L})}|$. The definition of $\theta_k^{(\text{FL})}$ implies that $|\theta_k^{(\text{FL})}| \leq \max_{m\in[M]}|\theta_m^{(\text{FL})}|$ for all $k \in [M]$. Since $\theta_m^{(\text{L})}$'s are independent Gaussian with variance bounded by $c_v$, we have $\max_{m\in[M]}|\theta_m^{(\text{FL})}| = O_p(\sqrt{\log M})$. Thus, we may choose

$$c_\theta = O_p(\sqrt{\log M}). \tag{39}$$

Taking (34) and (37) into (30), we obtain

$$|\zeta_m^t| \leq q \max_{m\in[M]} \left| \frac{\sum_{k:\,k\neq m} w_k \zeta_k^{t-1}}{\sum_{k:\,k\neq m} w_k} \right| + |D| \leq q \max_{m\in[M]} |\zeta_m^{t-1}| + \frac{|D| + 2vc_\theta}{1-v}, \quad \forall m \in [M]. \tag{40}$$

It follows from (40) that

$$\max_{m\in[M]} |\zeta_m^t| \leq q^t \max_{m\in[M]} |\zeta_m^0| + \frac{|D| + 2vc_\theta}{1-v}. \tag{41}$$

Next, we bound $|D|$. Recall that $\mathbb{E}(\theta_i^{(\text{FL})}) = \theta_0$ and $\text{Cov}(\theta_i^{(\text{FL})}, \theta_j^{(\text{FL})}) = 0$ for all $i, j \in [M]$ and $i \neq j$. Since

$$D = \frac{\sum_m w_m(\theta_m^{(\text{FL})} - \theta_m^{(\text{L})})}{\sum_m w_m}, \tag{42}$$

$$\theta_m^{(\text{FL})} - \theta_m^{(\text{L})} = \frac{\sum_{k:\,k\neq m} w_k \theta_k^{(\text{L})}}{\sigma_m^{-2} + \sum_{k:\,k\neq m} w_k} - \frac{\sum_{k:\,k\neq m} w_k}{\sigma_m^{-2} + \sum_{k:\,k\neq m} w_k}\theta_m^{(\text{L})} = \frac{\sum_{k:\,k\neq m} w_k(\theta_k^{(\text{L})} - \theta_m^{(\text{L})})}{\sigma_m^{-2} + \sum_{k:\,k\neq m} w_k} \tag{43}$$

$$\mathbb{V}(\theta_m^{(\text{FL})} - \theta_m^{(\text{L})}) \leq 2\mathbb{V}\left( \frac{\sum_{k:\,k\neq m} w_k \theta_k^{(\text{L})}}{\sigma_m^{-2} + \sum_{k:\,k\neq m} w_k} \right) + 2\mathbb{V}\left( \frac{\sum_{k:\,k\neq m} w_k}{\sigma_m^{-2} + \sum_{k:\,k\neq m} w_k}\theta_m^{(\text{L})} \right) \leq 4\max_k \mathbb{V}(\theta_k^{(\text{L})}) = 4V, \tag{44}$$

we have $\mathbb{E}(D) = 0$, and

$$\mathbb{V}(D) = \frac{\sum_m w_m^2 \mathbb{V}(\theta_m^{(\text{FL})} - \theta_m^{(\text{L})})}{(\sum_m w_m)^2} + \frac{\sum_{i,j:\, i \neq j} w_i w_j \text{Cov}(\theta_i^{(\text{FL})} - \theta_i^{(\text{L})}, \theta_j^{(\text{FL})} - \theta_j^{(\text{L})})}{(\sum_m w_m)^2} \tag{45}$$

$$\leq \frac{4V \sum_m w_m^2}{(\sum_m w_m)^2} + \sum_{i,j:\, i \neq j} \frac{w_i w_j \text{Cov}(\sum_{k:\, k \neq i} w_k(\theta_k^{(\text{L})} - \theta_i^{(\text{L})}), \sum_{k:\, k \neq j} w_k(\theta_k^{(\text{L})} - \theta_j^{(\text{L})}))}{(\sum_m w_m)^2 (\sigma_i^{-2} + \sum_{k:\, k \neq i} w_k)(\sigma_j^{-2} + \sum_{k:\, k \neq j} w_k)} \tag{46}$$

$$= \frac{4V \sum_m w_m^2}{(\sum_m w_m)^2} + \sum_{i,j:\, i \neq j} w_i w_j \frac{\sum_{k:\, k \neq i,j} w_k^2 \mathbb{V}(\theta_k^{(\text{L})}) - \left( w_j \mathbb{V}(\theta_j^{(\text{L})}) \sum_{k:\, k \neq j} w_k + w_i \mathbb{V}(\theta_i^{(\text{L})}) \sum_{k:\, k \neq i} w_k \right)}{(\sum_m w_m)^2 (\sigma_i^{-2} + \sum_{k:\, k \neq i} w_k)(\sigma_j^{-2} + \sum_{k:\, k \neq j} w_k)} \tag{47}$$

$$\leq \frac{4V \sum_m w_m^2}{(\sum_m w_m)^2} + \sum_{i,j,k:\, i \neq j, k \neq i, k \neq j} \frac{w_i w_j w_k^2 V}{(\sum_m w_m)^2 (\sigma_i^{-2} + \sum_{k:\, k \neq i} w_k)(\sigma_j^{-2} + \sum_{k:\, k \neq j} w_k)} \tag{48}$$

$$\leq \frac{4V}{M} \frac{w_{\max}^2}{w_{\min}^2} + \frac{MV w_{\max}^4}{w_{\min}^2 (\min_{m \in [M]} \sigma_m^{-2} + (M-1)w_{\min})^2} \tag{49}$$

$$\leq \frac{4V}{M} \frac{w_{\max}^2}{w_{\min}^2} + \frac{MV}{(M-1)^2} \frac{w_{\max}^4}{w_{\min}^4}. \tag{50}$$

Therefore, from (24), we have $\mathbb{V}(D) = O(M^{-1})$ as $M \to \infty$. By the Markov inequality, we further have

$$|D| = O_p(M^{-1/2}). \tag{51}$$

Consequently, taking equations (25), (39), and (51) into (41), we obtain

$$\max_{m \in [M]} |\zeta_m^t| = c_3 \cdot q^t + O_p(M^{-1/2}) + O(M^{-1}) = C \cdot q^t + O_p(M^{-1/2}) \text{ as } M \to \infty, \tag{52}$$

where $C$ is the constant upper bound of $\max_{m \in [M]} |\zeta_m^0|$ (by the assumption). This proves the convergence of each client's personalized model.

For the server model denoted by $\theta^t$, since

$$\theta^t = \frac{\sum_k w_k \theta_k^t}{\sum_k w_k} = \frac{\sum_k w_k(\theta_k^t - \theta_k^{(\text{FL})})}{\sum_k w_k} + \frac{\sum_k w_k(\theta_k^{(\text{FL})} - \theta_k^{(\text{L})})}{\sum_k w_k} + \frac{\sum_k w_k \theta_k^{(\text{L})}}{\sum_k w_k} \tag{53}$$

$$= \frac{\sum_k w_k \zeta_k}{\sum_k w_k} + D + \theta^{(\text{G})}, \tag{54}$$

we have $|\theta^t - \theta^{(\text{G})}| \leq C \cdot q^t + O_p(M^{-1/2})$. This concludes the proof.