# OpenReview forum: "Self-Aware Personalized Federated Learning"
_NeurIPS.cc/2022/Conference — NeurIPS 2022 Accept_

### Official Review · Reviewer_rRk4 · 2022-07-11

**Rating:** 3
**Confidence:** 3
**Soundness:** 2 fair
**Presentation:** 2 fair
**Contribution:** 2 fair

**Summary:**

The authors present a self aware personalized FL system which uses Bayesian hierarchical modeling to balance training of local versus global models. This helps to quantify inter and intra client uncertainty quantification. The authors present both theoretical justification and empirical analysis of the proposed technique - Self-FL.

**Questions:**

- The motivation for using Bayesian hierarchical modeling for personalization needs to be better explained.

-  How exactly is personalization quantified and how is the notion of "personalization" different from learning parallel algorithms where local models are updated by information from global models?

Minor comments
Section 2.1. to bake prior knowledge -> to take prior knowledge



**Limitations:**

- The use of local versus global training  is extensively used in literature. It is unclear how this idea is different from the notion of "personalization" introduced here. Infact, the basic premise seems to be the construction of a generative model at the client side and update its parameters using a global counterpart (assuming a generative model for global training).


**Strengths And Weaknesses:**

Strengths
- The authors identify an important problem -- that of personalizing federated learning problems
- The paper is well written
- A reasonably well written Related Work and Experimental Studies section

Weakness
- The problem of tuning local versus global models has been studied extensively in literature and therefore the novelty of the proposed approach is questionable.

---

> ### Author Response · Authors · 2022-07-27
> **Responses to Reviewer rRk4**
>
> Thank you for your time and comments. Our review responses are summarized below.
>
> **Reviewer's Comment:** ‘The problem of tuning local versus global models has been studied extensively in literature and therefore the novelty of the proposed approach is questionable’.
>
> **Answer:** There might be some misunderstanding regarding the novelty of this work. The introduction and Appendix Section C explained the difference between our proposed Self-FL framework and previous FL methods. We also clarified the novelty of our uncertainty-driven personalization FL scheme in Section 3 of the main body. We hope you can re-evaluate our novelty accordingly and kindly give details about any concerns.
>
> **Reviewer's Comment:** The motivation for using Bayesian hierarchical modeling for personalization needs to be better explained.
>
> **Answer:** We showed how personalization (i.e., learning local models) and Bayesian hierarchical modeling are tightly connected in the motivating case where the data distributions follow two-level Gaussian in Section 2.1.
> For real-world FL systems, let us consider the task of speech recognition (SR) performed by home assistants (e.g., Alexa) as an instance. In this exemplar application, the general device class has a `root’ SR model ($\theta_0$). This general device class can be divided into multiple device cohorts using geographical information where each cohort hosts its local SR model ($\theta_m$). The local model is obtained using the knowledge of local observations ($z_m$) and the root model  ($\theta_0$). We observe that the information flow from the bottom level (local observations $z_m$) to the middle level (local models $\theta_m$) and finally to the top level (root model $\theta_0$) resembles the structure of hierarchical modeling. Therefore, we are motivated to use Bayesian hierarchical modeling for learning personalized local models ($\theta_m$).
>
> **Reviewer's Comment:** How exactly is personalization quantified?
>
> **Answer:** We quantify `personalization’ with the test performance of a client’s local model on its local data distribution. We provided the definition of the client-level test accuracy that aggregates the local performance across all clients in Section 4.2 (page 7) and use it as the metric to quantify the personalization performance of FL algorithms. The corresponding results are shown in Figure 1.
>
> **Reviewer's Comment:** How is the notion of "personalization" different from learning parallel algorithms where local models are updated by information from global models?
>
> **Answer:** We are not sure what you meant by ‘learning parallel algorithms where local models are updated by information from global models’ since it seems to be the general setup of federated learning. We discussed existing FL works in Appendix Section C and highlighted that the goal of general FL is to obtain a well-performing global model. However, the goal of personalized FL is to obtain a set of personalized models that perform well on the local data of clients. We designed an FL method (see Alg.1) that is particularly tailored for personalization purposes.
>
> **Reviewer's Comment:** In fact, the basic premise seems to be the construction of a generative model at the client side and update its parameters using a global counterpart (assuming a generative model for global training).
>
> **Answer:** There seems to be a significant misunderstanding about our paper from this comment. Our proposed framework does not involve the construction of generative models on the client side and does not assume a generative model for global training. We started with a two-level Gaussian model in Section 2 as a motivating example and then developed further technical insights and practical solutions for general learning problems in later sections.

---

> ### Author Response · Authors · 2022-08-04
> **Followup Response to Reviewer rRk4**
>
> Dear Reviewer rRk4,
>
> We would like to thank you again for the time you dedicated to reviewing our paper. We believe that we have addressed your concerns. Since the end of discussion period is getting close and we have not heard back from you yet, we would appreciate it if you kindly let us know of any other concerns you may have, and if we can be of any further assistance in clarifying any other issues.
>
> Thanks a lot again, and with sincerest best wishes
>
> Authors

---

### Official Review · Reviewer_jvSV · 2022-07-11

**Rating:** 7
**Confidence:** 3
**Soundness:** 3 good
**Presentation:** 3 good
**Contribution:** 3 good

**Summary:**

The paper is on personalized federated learning, where the critical challenge is in balance of local and global model tuning. More specially, the goal of global model tuning and local personalized model tuning are not fully aligned.  Toward this problem, the authors develop a self-aware personalized FL method to balance the personal model and global model update based on the inter-client and intra-client uncertainty quantification. The paper recognized and clearly defined critical research questions. To answer the questions, the authors formulate  personalization from a  hierarchical model-based perspective and propose Self-FL, an active personalized FL solution that guides local training and global aggregation via inter- and intra-client uncertainty quantification. The experimental results confirm the effectiveness of the proposed method from task performance and training stableness.

**Questions:**

One of the challenges in personalization is how to achieve good performance while clients do not have enough data and prevent overfitting.  Could authors provide some intuitions to explain how the model handle the case when the labeled data is limited on some clients?

**Limitations:**

The authors propose to characterize the performance by weighted accuracy. Even though this is a good way to show the performance, the detailed performance distribution is also suggested. For example, a violin plot could depict the distribution of performance.
Another suggestion is to uncover the relationship between performance and other factors including data size and class distribution on client sides.

**Strengths And Weaknesses:**

The authors clearly defined the challenging questions and work on the fundamental research questions. The proposed methods are derived  in a principal way with theory. The performance is promising and experiments are pretty comprehensive. The personalization naturally puts more requirements on evaluation and analysis due to involving performance evaluation across clients. I include my suggestions in the following sections and encourage authors to provide more analysis to uncover personalization challenges in term of data dimension.

---

> ### Author Response · Authors · 2022-07-27
> **Responses to Reviewer jvSV**
>
> Thank you for your positive feedback and constructive comments. Our review responses are summarized below.
>
> **Reviewer's Comment:** One of the challenges in personalization is how to achieve good performance while clients do not have enough data and prevent overfitting. Could authors provide some intuitions to explain how the model handle the case when the labeled data is limited on some clients?
>
> **Answer:** Here is our intuition regarding the case when a client does not have sufficient labeled data. The client’s intra-client uncertainty value $\sigma_m^2$ tends to be large in this case and the local training step $l$ computed via Equation (8) will be small. Therefore, Self-FL will suggest a client with limited data observations train only a few epochs on their local dataset, which will alleviate overfitting.
>
> **Reviewer's Comment:** The authors propose to characterize the performance by weighted accuracy. Even though this is a good way to show the performance, the detailed performance distribution is also suggested.
>
> **Answer:** We appreciate the suggestion of detailed performance distribution in addition to the weighted accuracy metric. In that direction, we have provided a fine-grained view of Self-FL’s user-level performance in Table 1 of the paper, where the clients with the most amount of data and the clients with the worst local performance are reported. In general, we observed from our experiments that the user-level test accuracy distribution under Self-FL is shifted toward the right side of its counterpart under baseline approaches. We will add other figures to show more details of the performance distribution in the revised paper, such as the suggested violin plot.
>
> **Reviewer's Comment:** Another suggestion is to uncover the relationship between performance and other factors including data size and class distribution on client sides.
>
> **Answer:** Thank you for the suggestion. We will add ablation studies to reveal the relation between Self-FL’s performance and various factors such as data dimensions, local data sizes, and class distributions of clients in the revised paper.

---

### Official Review · Reviewer_b7BG · 2022-07-12

**Rating:** 7
**Confidence:** 3
**Soundness:** 3 good
**Presentation:** 3 good
**Contribution:** 3 good

**Summary:**

In this paper, the authors study an important problem in federated learning, i.e., personalized federated learning, where the goal is to balance the training of the local model of each client and the global model shared by all the clients. In particular, the authors do not follow previous works on model fine-tuning in each local client and size-based weighting in global aggregation, but propose to address the balancing challenge from a statistical uncertainty perspective. Specifically, the authors design a novel solution called Self-FL, which uses uncertainty-driven approaches for the training of the local models and the global model. Empirical results on some well-known datasets show the effectiveness of the proposed Self-FL.

**Questions:**

1 Can the proposed approach be applied to other tasks, e.g., classification/recognition with some auxiliary data such as knowledge graph, federated recommendation?

**Limitations:**

The authors do not include sufficient discussions about the communication and computational cost, which are suggested to be included.

**Strengths And Weaknesses:**

Strengths:

1 The authors address an important challenge in personalized federated learning, i.e., balancing of the training of each local model and the global model.

2 The authors propose a novel perspective for the studied problem and design some novel uncertainty-driven approaches.

Weakness:

1 The authors may include more discussions and quantitative analysis about the communication cost.

---

> ### Author Response · Authors · 2022-07-27
> **Responses to Reviewer b7BG**
>
> Thank you for your positive feedback and constructive comments. Our review responses are summarized below.
>
> **Reviewer's comment:** The authors may include more discussions and quantitative analysis about the communication cost.
>
> **Answer:** The proposed Self-FL framework (described in Alg.1) requires the selected clients to send their updated local models $\theta_m^t$ and the intra-client uncertainty values $\sigma_m^2$ to the server. After uncertainty-aware model aggregation, the server distributes the updated global model and the inter-client uncertainty $\sigma_0^2$ to the clients. Therefore, the communication cost of Self-FL is the same level as the standard FedAvg except for the additional cost of transmitting the uncertainty values (which are scalars).
>
> **Reviewer's comment:** Can the proposed approach be applied to other tasks, e.g., classification/recognition with some auxiliary data such as knowledge graph, federated recommendation?
>
> **Answer:** We believe the proposed personalization method to balance local and global training in FL applies to other tasks involving auxiliary data since our method does not depend on a particular formulation of the empirical risk. We will investigate the performance of Self-FL on other applications suggested by the reviewer in future work.

---

### Meta-Review · Area_Chair_C8g9 · 2022-08-26

**Recommendation:** Accept
**Confidence:** Less certain

**Metareview:**

In this submission, the authors study personalized federated learning and propose a self-aware personalized method to address the balancing challenge from the perspective of uncertainty quantification. This problem is interesting and important (as pointed out by b7BG and rRk4), and the proposed method is derived in a principal way from theory (as pointed out by jvSV) and useful for applications. I recommend accepting this submission.

The authors can include some discussion about the communication and computation cost (as suggested by b7BG), and add an ablation study to show the effect of other factors (as suggested by jvSV), to make this submission better.
Hope that the suggestions from all the reviewers and the discussion between reviewers and authors can make this submission a better on.



**Award:**

No

---

### Decision · Program_Chairs · 2022-09-14

Accept